https://doi.org/10.1038/s42003-023-05397-7　　**OPEN**
# Mendelian randomization and colocalization analyses reveal an association between short sleep duration or morning chronotype and altered leukocyte telomere length

Jingyi Hu [1,5✉], Jiawen Lu[2,5], Qiuhan Lu[2], Weipin Weng[3], Zixuan Guan[4] & Zhenqian Wang [2✉]

Observational studies suggest certain sleep traits are associated with telomere length, but the causal nature of these associations is unclear. The study aimed to determine the causal associations between 11 sleep-related traits and leukocyte telomere length (LTL) through two-sample Mendelian randomization and colocalization analyses using the summary statistics from large-scale genome-wide association studies. Univariable Mendelian randomization indicates that genetically determined short sleep is associated with decreased LTL, while morning chronotype is associated with increased LTL. Multivariable Mendelian randomization further supports the findings and colocalization analysis identifies shared common genetic variants for these two associations. No genetic evidence is observed for associations between other sleep-related traits and LTL. Sensitivity MR methods, reverse MR and re-running MR after removing potential pleiotropic genetic variants enhance the robustness of the results. These findings indicate that prioritizing morning chronotype and avoiding short sleep is beneficial for attenuating telomere attrition. Consequently, addressing sleep duration and chronotype could serve as practical intervention strategies.

[1] National Clinical Research Center for Metabolic Diseases, Key Laboratory of Diabetes Immunology, Ministry of Education, Department of Metabolism and Endocrinology, The Second Xiangya Hospital of Central South University, Changsha, Hunan, China. [2] School of Public Health (Shenzhen), Sun Yat-sen University, Shenzhen, Guangdong 518107, China. [3] Department of Neurology, Center for Cognitive Neurology, Fujian Medical University Union Hospital, Fuzhou, Fujian 350001, China. [4] Chongchuan District Center for Disease Control and Prevention, Nantong, Jiangsu 226001, China. [5] These authors contributed equally: Jingyi Hu, Jiawen Lu. ✉email: hujingyi0066@csu.edu.cn; wangzhq68@mail2.sysu.edu.cn

Telomeres are intricate and dynamic structures located at the termini of chromosomes, composed of repetitive nucleotide sequences that form a "cap structure"[1]. In human beings, telomere length is preserved through the activity of telomerase, an enzyme that adds telomere repeats to the ends of chromosomes. This process offsets the gradual loss of telomeric DNA with each cell division, ensuring the maintenance of the genome's stability and integrity. Despite this protective mechanism, telomeres naturally shorten over time. This phenomenon, known as telomere attrition, has been linked to several age-related diseases, including cancer, cardiovascular disease, and neurodegenerative diseases[2]. Consequently, telomeres have emerged as a critical factor in understanding the molecular basis of aging and age-related pathologies, with leukocyte telomere length (LTL) being widely studied as a biomarker of aging and disease.

Sleep is an essential activity of human physiological regulation. Due to changes in lifestyle and circadian rhythm, sleep has evolved into many types, such as evening or morning chronotypes. Extensive evidence has demonstrated a noteworthy correlation between sleep traits and LTL. For example, cross-sectional and case-control studies have linked sleep deprivation, sleep disruptions, and evening chronotype to shortened LTL[3,4]. Likewise, evidence suggests that inadequate sleep quality, notably extended sleep latency, insufficient sleep duration (<5 h), and decreased sleep efficiency (<65%), could be contributing factors to telomere shortening[5]. However, the assessment of sleep traits in most studies predominantly relied on self-reported data, raising concerns about its potential limitations in accurately capturing actual sleep patterns and susceptibility to recall bias. Moreover, given the nature of observational studies, they may not be sufficiently robust in establishing cause-and-effect relationships. To address this issue, Mendelian randomization (MR) studies are commonly used to infer causality between an exposure and an outcome using genetic variants randomly allocated at conception[6]. Nonetheless, the only MR study reported conflicting results, showing no causal relationship between insomnia and LTL[7]. Thus, additional evidence is imperative to disentangle the correlation between sleep and LTL, with particular emphasis on different types of sleep patterns.

In this study, we aimed to comprehensively investigate the causal associations between sleep traits and LTL by conducting two-sample MR analyses on a set of seven self-reported and four accelerometer-based sleep traits. We hypothesized that a shared molecular mechanism might be responsible for the genetic correlation between sleep traits and TL, and we further explored this possibility using colocalization analyses. Understanding the causal relationships between sleep traits and LTL is of paramount importance, as it bears substantial implications for unraveling the potential consequences of sleep patterns on cellular senescence.

## Results

Detailed information on the genetic IVs for sleep-related traits after LD clumping and harmonization is shown in Supplementary Data 1. The mean $F$-statistics for sleep traits ranged from 37.59 to 59.37, indicating little chance of a weak-instrument bias (Supplementary Data 2).

**UVMR analyses of each sleep-related trait on LTL**. One-unit higher log odds of self-reported short sleep duration decreased 0.315 standard derivation (SD) of LTL ($\beta$ [95% CI]: −0.315 [−0.451, −0.178]; FDR-corrected $P < 0.001$; Fig. 1). The MR result of self-reported short sleep duration indicated that per doubling of prevalence decreased by 0.218 SD (multiply the causal estimate by 0.693) of LTL[8]. The IVW method showed genetically predicted morning chronotype was associated with the longer LTL (0.016

[0.004, 0.028]; FDR-corrected $P = 0.049$) compared with the evening chronotype (Fig. 1). There was no evidence for causal relationships between other sleep-related traits and LTL in IVW method and other sensitivity MR methods (all $P > 0.05$; Fig. 1 and Supplementary Data 3). No horizontal pleiotropy was detected using MR-Egger regression (Supplementary Data 3). Although the MR-pleiotropy residual sum and outliers (MR-PRESSO) method detected horizontal pleiotropy, the results for self-reported short sleep duration (−0.315 [−0.451, −0.178]; $P < 0.001$), morning chronotype (0.016 [0.001, 0.031]; $P = 0.032$), and other sleep traits were not substantially altered (all $P > 0.05$) after outlier-correction (Supplementary Data 3). Scatter plots depicting the genetic associations with LTL against the genetic associations with the sleep-related traits were provided (Supplementary Fig. 1). In the leave-one-out analysis, no apparent outlying SNPs were observed, and the results were not influenced by any outlier and clustered closely around the expected value of estimation (Fig. 2 and Supplementary Figs. 2 and 3). Radial MR analyses identified one to twenty outliers for these UVMR analyses of 11 sleep-related traits on LTL (Supplementary Figs. 4 and 5). After excluding the outliers detected by radial MR analyses, the results did not change markedly (Supplementary Data 4). Using the MR-Steiger filtering method, the exposure of self-reported sleep duration, self-reported short sleep, and daytime napping removed SNPs suggestive of reverse causation. Subsequent MR analyses excluding these SNPs showed similar results to the primary analysis (Supplementary Data 5).

Additionally, we scanned each genetic IV in the PheWAS to examine whether it was associated with BMI, Townsend deprivation index, physical activity, smoking, alcohol consumption, and waist-to-hip ratio. If an SNP is associated with secondary phenotypes (which is pleiotropic SNP), we removed it and repeated the UVMR analyses. The number of removed IVs and remaining IVs are shown in Supplementary Data 6, and mean $F$-statistics for all sleep traits proxied by remaining IVs are more than 10 (Supplementary Data 2). After the exclusion of the potential pleiotropic SNPs, we found the relationships between genetically determined self-reported short sleep duration (−0.292 [−0.435, −0.149]; $P < 0.001$) and morningness (0.014 [0.002, 0.027]; $P = 0.024$) with LTL remained consistent both in direction and significance using IVW method (Supplementary Data 6). We also observed no evidence for the causal associations between other sleep traits and LTL, which was consistent with the main analyses (Supplementary Data 6).

**Reverse MR analyses of LTL on self-reported short sleep duration and chronotype**. To ensure that the causal direction was from chronotype and short sleep duration to LTL, rather than vice versa, we conducted MR analysis on LTL with chronotype and short sleep duration. Further information on the genetic IVs for LTL after LD clumping and harmonization was listed in Supplementary Data 7 (mean $F$-statistics = 108.69 in Supplementary Data 2). IVW, MR-Egger, weighted median, and MR-PRESSO showed no causal evidence from LTL to chronotype and short sleep duration (all $P$-values > 0.05; Supplementary Data 8).

**MVMR analyses of self-reported short sleep duration and chronotype on LTL**. To further examine the role of self-reported short sleep duration and chronotype on LTL, we perform MVMR to test whether self-reported short sleep duration and chronotype could be causally associated with LTL after adjusting for the genetic association with potential confounders, such as smoking, alcohol consumption, BMI, and other sleep-related traits. MVMR-IVW method showed the inverse direct effect of self-reported short sleep duration on LTL (−0.159 [−0.310, −0.009];

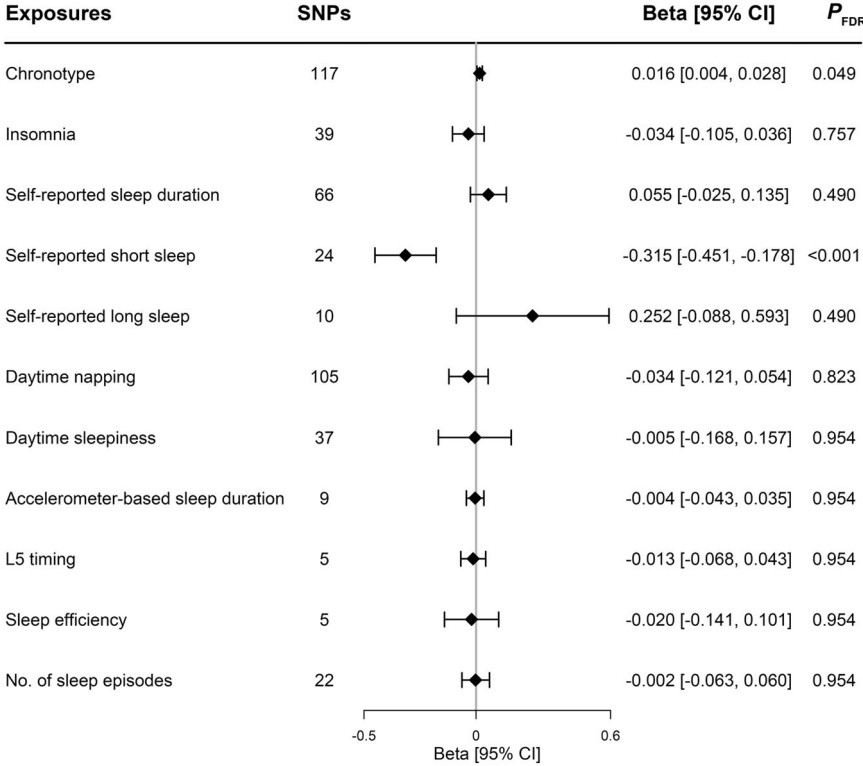

**Fig. 1 The effect of genetically determined sleep-related traits on LTL using UVMR.** Abbreviations: LTL leukocyte telomere length, SNPs single-nucleotide polymorphisms, L5 timing least active 5 h timing, $P_{FDR}$ FDR-corrected $P$-value. The error bars indicated the 95% confidence interval corresponding to the estimates of 11 sleep-related traits on LTL.

$P = 0.038$) after adjusting for the effect of BMI, smoking, and alcohol consumption (Fig. 3). After adjusting for the effect of self-reported sleep duration, insomnia, BMI, smoking, and alcohol consumption, we observed that MVMR results supported the findings of a positive effect of morning chronotype on LTL (0.017 [0.001, 0.033]; $P = 0.048$; Fig. 3). The MVMR-Egger method provided consistent results and suggested the absence of horizontal pleiotropy (Supplementary Data 9 and 10). These findings remained consistent in the MVMR-LASSO, MVMR-PRESSO, and MVMR-Q(het) methods (Supplementary Data 9 and 10).

**Colocalization analyses of chronotype and short sleep duration with LTL.** We totally performed colocalization analysis in 117 and 24 gene regions for the association between chronotype and LTL and the association between short sleep duration and LTL, respectively (Supplementary Data 11). Colocalization analysis revealed that short sleep duration and LTL association had a 97.34% PP.H4 of sharing a causal variant within the gene region (±200 kb) of rs2517827 (Fig. 4a and Supplementary Data 11). There was evidence of an association between chronotype and LTL within the gene region of rs11712056 (PP.H4 = 83.88%; Fig. 4b). Moreover, rs2517827 and rs11712056 were considered as the most likely shared causal variant for the two regions showing evidence for colocalization (Fig. 4).

**Evaluation of the impact of overlap.** Because there are partially overlapping sets of participants between the two samples, we investigate whether the degree of bias introduced by sample overlap impacted our findings. The degree of overlap between individuals of sleep-related traits and individuals of LTL ranged from 17.88% to 95.86% (Supplementary Data 12) and the lower bound of one-sided 95% CI for the $F$ parameter ranged from 33.54 to 44.06 (Supplementary Data 2). The type 1 error rate due

to the sample overlap between sleep-related traits and LTL was controlled under 0.05, and the biases were minimal (Supplementary Data 12). Additionally, the MRlap analyses showed concordant results with primary results (Supplementary Data 13). Hence, despite the considerable overlap between the two samples, considerable weak instrument bias would not be expected.

**Discussion**
In the current study, we conducted two-sample MR analyses to comprehensively explore the causal associations between 11 sleep-related phenotypes and LTL. Our results revealed that genetically determined self-reported short sleep duration was directly associated with shortened LTL, while genetic predisposition to morning chronotype was suggestively associated with increased LTL. Reverse MR indicated that genetically predicted LTL does not exert an effect on chronotype and short sleep duration. Further evidence supporting the potential causal associations of short sleep duration and chronotype with LTL was observed in sensitivity analyses, reinforcing our findings. Finally, the colocalization analysis revealed that the genetic variant rs2517827 was shared between short sleep duration and LTL, while rs11712056 was shared between chronotype and LTL, supporting the possibility of a shared genetic basis between these traits.

Short sleep duration exerts a substantial influence on an individual's overall health and is strongly linked with increased morbidity and mortality[9]. Numerous studies have reported that short sleep duration was associated with shorter telomere length[10,11]. Consistent with these observational studies, our study findings demonstrated a causal association between short sleep duration and telomere attrition. By contrast, no causal relationships were observed for overall sleep duration and long sleep duration, suggesting no linear causality between sleep duration and LTL exists. Several biological mechanisms have been suggested as a potential link between short sleep

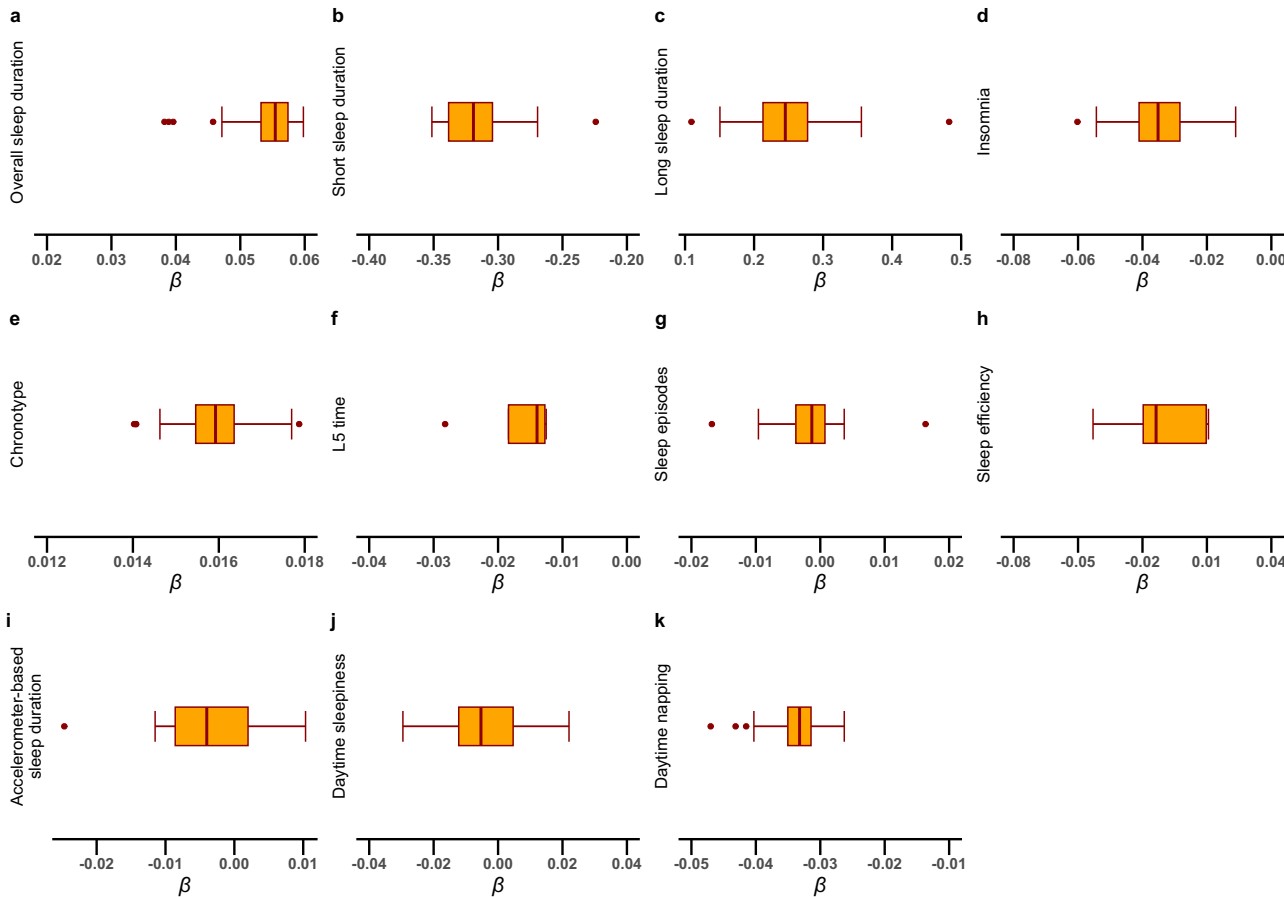

**Fig. 2 Sensitivity analysis leaving one SNP out at a time for the association between sleep-related traits and LTL.** Each boxplot represents the centralized tendency of effect sizes ($\beta$ coefficients) of **a** overall sleep duration, **b** short sleep duration, **c** long sleep duration, **d** insomnia, **e** chronotype, **f** L5 time, **g** sleep episodes, **h** sleep efficiency, **i** accelerometer-based sleep duration, **j** daytime sleepiness, **k** daytime napping on LTL based on the results of leave-one-out analysis where we excluded one SNP at a time and performed IVW using the remaining SNPs. The line in the box indicates the median based on the results of the leave-one-out analysis. The left line of the box represents the first quartile (Q1), which is the 25th percentile. The right line of the box represents the third quartile (Q3), which is the 75th percentile. The width of the box is the interquartile range (IQR). The points outside the whiskers are potential outlying SNPs.

duration and shorter telomere length. Telomere length is regulated by inflammation, oxidative stress, sympathetic tone, and endocrine level[12–14]. Short sleep duration will cause sleep deprivation and long-term sleep debt, which will affect the physiological regulation of the body, resulting in the activation of an inflammatory cascade reaction, increasing levels of oxidative stress[12,15]. The consequent increase in levels of inflammatory factors and reactive oxygen species can damage telomere structure and accelerate telomere wear. Moreover, short sleep duration can activate neuroendocrine stress systems and promote the release of cortisol hormone[16]. Cortisol, in turn, can reduce telomerase activity and prevent telomere repair from extending[17,18]. However, once the body's sleep requirements are met, further variations in sleep duration no longer appear to have an essential impact on telomere length.

Chronotype changes are a manifestation of biological rhythm disorders that are influenced by social and environmental factors. An increasing number of individuals have shifted from morning to evening chronotypes, exhibiting delayed circadian rhythms. In this study, we observed that there was a suggestively causal relationship between chronotype and telomere length, with morning chronotypes exhibiting longer telomeres than evening chronotypes. This correlation was consistent with previous research results[4]. People with evening chronotype typically go to bed late and have difficulty waking up in the morning, which also causes sleep deprivation and chronic sleep debt[19]. They may additionally exhibit impaired social

and occupational function, leading to higher stress levels[4,20]. In this case, even a single night of sleep deprivation can prominently increase inflammatory activity in healthy individuals[21]. Furthermore, the effect of delayed circadian rhythm on telomeres may be associated with inflammation increasing and disorder of cell circadian metabolism, ultimately leading to cell damage and accelerating cell aging[4]. Studies have also linked circadian rhythm disorders with mental health issues, such as bipolar disorder, which has been associated with shorter telomere length in patients[22]. Therefore, our study highlights the importance of maintaining a regular sleep–wake schedule and a morning-oriented lifestyle to promote healthy aging.

Through colocalization analysis, we identified that the genetic variant rs2517827 was shared between short sleep duration and LTL, while rs11712056 was shared between chronotype and LTL. The SNP rs2517827 acted as a *cis*-eQTL that affects *HLA-G* expression, which is located on the telomeric end of the MHC genomic region[23]. In addition, Chen et al. reported associations between rs2517827 and various hematological indicators, such as white blood cell count, lymphocyte count, and monocyte count, which provided a potential molecular mechanism linking this genetic variation with telomere length[24]. Another causal genetic variant, rs11712056, is important for predicting *CAMKV* expression[25]. *CAMKV* is involved in regulating calcium signaling, synaptic plasticity, and memory formation in the brain, and

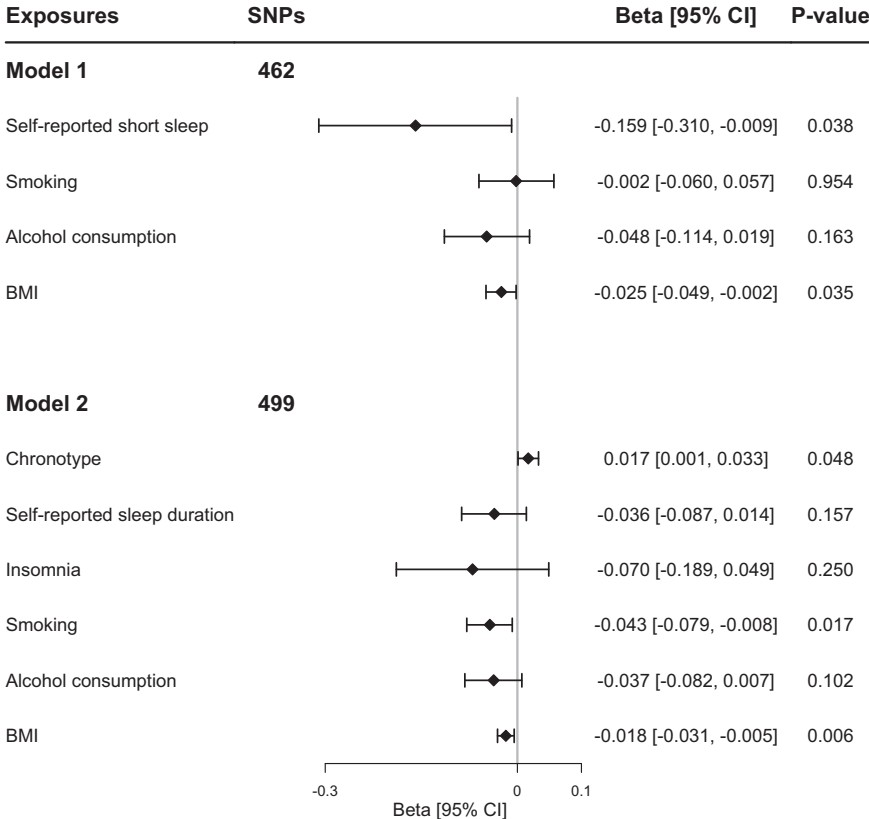

**Fig. 3 The direct effect of genetically determined self-reported short sleep duration and morning chronotype on LTL using MVMR adjusted for smoking, alcohol consumption, BMI, and other sleep traits.** Abbreviations: BMI body mass index, LTL leukocyte telomere length, SNPs single-nucleotide polymorphisms. The reference group of self-reported short sleep (≤6 h) is sleep duration between ≥7 h and <9 h every 24 h. The reference group of the chronotype is the evening preference chronotype. The error bars indicated the 95% confidence interval corresponding to the estimates of these exposures on LTL.

alterations in these processes have been linked to various age-related diseases such as Alzheimer's or Parkinson's disease[26].

Apart from the aforementioned findings, our study did not yield evidence to support a causal relationship between other sleep traits and LTL. However, the association between such as insomnia, daytime sleepiness, daytime napping, and LTL remains equivocal, as previous studies have produced inconsistent results. For instance, a recent study showed that there was a causal relationship between telomere length and insomnia in women, but this connection did not hold for the whole study population[7]. Ding et al. also found that insomnia symptoms were related to telomere length in people over 55 years old, whereas the results were not statistically significant in participants aged 40–54[27]. Similarly, a study investigating the impact of maternal sleep apnea on fetal LTL found no notable difference in telomere length between the group with normal daytime sleepiness and the group with abnormal daytime sleepiness, while the mean telomere length was longer in the normal group compared to the abnormal group[28]. Thus, it is plausible that the conflicting findings in previous observational studies may be attributed to the influence of self-report bias and unmeasured confounding factors. In the current study, L5 timing was assessed using an accelerometer, which captures subtle body movements during sleep, thereby providing a more precise measurement of sleep timing compared to self-reported methods. L5 timing is frequently utilized to assess an individual's circadian rhythm. Despite the accelerated telomere shortening associated with impaired circadian rhythms, the causal relationship between L5 timing and LTL was not statistically significant. This may suggest that circadian rhythms influence telomere length through other indicators, such as the type of early or late sleep we mentioned

earlier. In addition, sleep frequency and sleep efficiency are commonly utilized as indicators of sleep quality, although their impact on telomere length remains inconclusive in light of prior research. Some studies reported that there is no correlation between sleep quality and telomere length[29–32], while other studies reported an association between poor sleep quality and short telomere length[33,34]. Poor sleep quality and low sleep efficiency were reported to be associated with faster longitudinal shortening of telomere length[5]. The results obtained from our study suggest that sleep frequency and sleep efficiency do not have a causal influence on telomere length. One conceivable explanation for this outcome could be the potential for bias in the correlation between subjectively reported sleep variables and telomere length, which underscores the need for the inclusion of multiple indicators to ensure a thorough causal assessment.

There are several distinct advantages to our study. We leveraged the most recent and comprehensive GWAS data available for sleep traits and LTL and thoroughly investigated a broad range of sleep-related traits in our study. Our study represents an innovative attempt to investigate whether sleep-related traits can cause the change in LTL and whether common genetic mechanisms underlie significant sleep traits and LTL, which was not been previously examined in MR studies. Results of all analyses consistently demonstrated a significant association between genetic predisposition to self-reported short sleep and morning chronotype with LTL. A colocalization study demonstrated associations of short sleep and chronotype with LTL shared common genetic causal variants in a given region, indicating shared genetic mechanisms may exist among them. Moreover, reverse MR provided no evidence of causal associations of LTL with chronotype and short sleep duration, ensuring

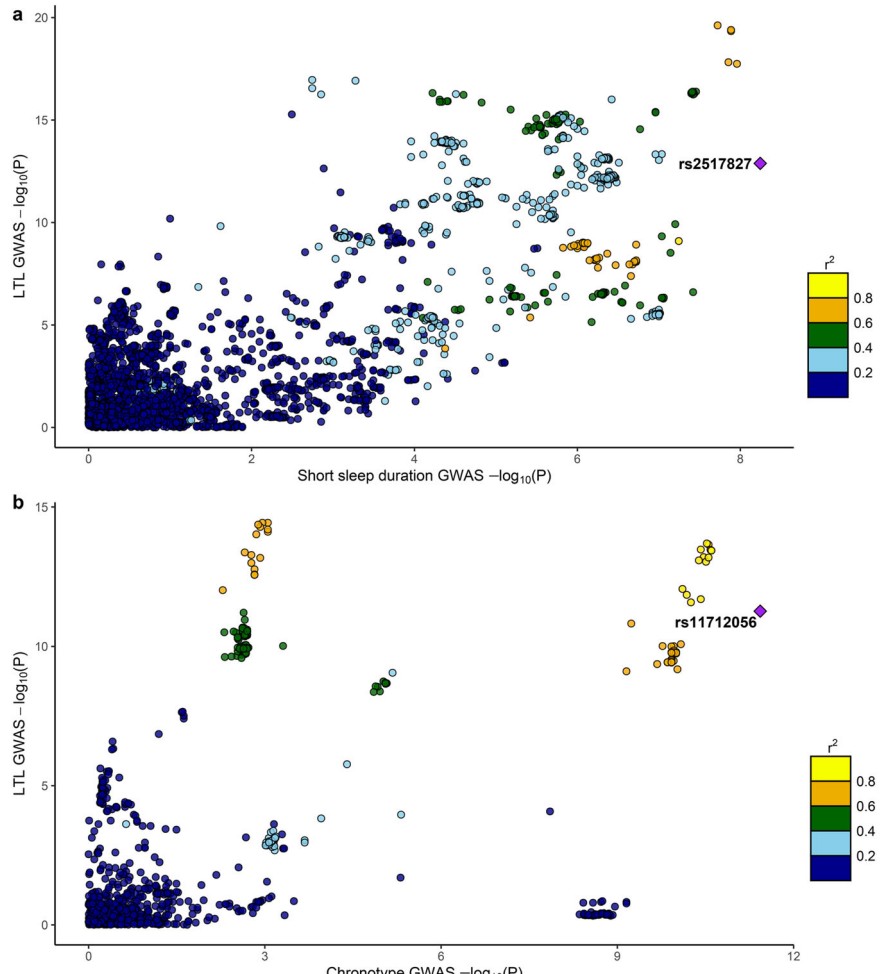

**Fig. 4 Locus comparing plots for the shared causal variant for the associations of short sleep duration and chronotype with leukocyte telomere length.**
**a** Colocalization analysis results for the association between short sleep duration and LTL in the gene region (Chr6:29632846-30032846), which is located within ±200 kb from rs2517827. In this region, rs2517827 is the lead variant identified in the GWAS of short sleep duration and is strongly correlated with the lead variant identified in the GWAS of LTL (LD $r^2 > 0.6$). **b** Colocalization analysis results for the association between chronotype and LTL in the gene region (Chr3:49714397-50114397), which is located within ±200 kb from rs11712056. In this region, rs11712056 is the lead variant identified in the GWAS of chronotype and is strongly correlated with the lead variant identified in the GWAS of LTL (LD $r^2 > 0.8$). Abbreviations: LTL leukocyte telomere length.

the causal direction was from chronotype and short sleep duration to LTL. These findings could potentially inform the development of targeted interventions aimed at ameliorating telomere attrition and promoting healthy aging.

Several limitations also should be considered when interpreting our findings. First, sample overlap of the current two-sample MR study due to both samples of sleep-related traits and LTL being from UK Biobank may lead to bias and type 1 error (false positive) inflation. For this, the type 1 error rate due to sample overlap between sleep-related traits and LTL was controlled under 0.05, and the MR estimates were well validated by using the MRlap method. Furthermore, a recent simulation study provided support for the rationality and validity of two-sample MR using overlapping samples in large cohorts[35]. Second, despite the fact that we cannot definitively exclude the potential for pleiotropic effects in our analyses, multiple sensitivity analyses under different assumptions were undertaken to assess the robustness of our results. Notably, these analyses yielded concordant conclusions, supporting the validity of our findings. Third, the present study solely consisted of individuals of European ancestry, thereby limiting the generalizability of our findings to other ethnic groups. Fourth, the causal associations between sleep-related traits and LTL were explored under linear assumption, and a non-linear effect may

exist that was not captured due to current data availability. Future research could explore potential non-linear associations using individual-level data. Finally, our measurement of telomere length was restricted to leukocytes, and it is unclear to what extent this measurement is representative of telomere length in other organs[36].

In the current study, we performed two-sample MR designs for 11 sleep-related traits and LTL and found evidence supporting a potential causal influence of short sleep duration and morning chronotype on LTL. Our findings provide evidence that sleep-related traits, particularly at short sleep duration and chronotype, are a causal determinant of telomere length. Furthermore, shared genetic mechanisms may underlie short sleep duration and LTL, as well as chronotype and LTL. Adherence to normal sleep duration and morning chronotype could have beneficial effects, as shortened telomeres have been proposed as crucial risk factors for various age-related diseases and short longevity.

## Methods

**Design overview.** Figure 5 describes the outline of the overall MR designs. Initially, we employed univariable MR (UVMR) analysis, utilizing single-nucleotide polymorphisms (SNPs) as genetic

instrumental variables (IVs) to proxy each sleep trait, to make causal inference of 11 sleep traits with LTL. Several sensitivity analyses were conducted to strengthen the robustness and reliability of the UVMR results. For sleep traits that showed significant results in the aforementioned analysis, we performed multivariable MR (MVMR) analyses to assess the direct effect of these sleep traits on LTL, with the adjustment of potential confounders (i.e., other sleep traits, drinking, smoking, and BMI). Finally, colocalization analysis was performed to assess whether significant sleep traits and LTL shared the same genetic causal variant in a given gene region.

**Data sources**. The current study included 11 sleep-related traits as exposures, comprising self-reported duration of overall sleep, short sleep (≤6 h), long sleep (≥9 h), insomnia, chronotype, daytime napping, daytime sleepiness, accelerometer-based sleep duration, least active 5 h timing (L5 timing), sleep efficiency, and the number of sleep episodes. The study outcome was the measurement of

LTL. Summary statistics for each trait were derived from recent large-scale European genome-wide association studies (GWASs). All studies cited in publicly available GWASs received approval from the respective ethical review boards, and all participants in these cited GWASs provided informed written consent. Detailed information on these GWASs is presented in Table 1.

**Exposures**. Overall sleep duration was evaluated through self-reporting by asking participants how many hours of sleep they typically get in a 24-h period (including naps), with only integer values accepted as an answer ($N = 446{,}118$)[37]. Additionally, binary variables were created to distinguish short sleep duration (≤6 h vs. 78 h) and long sleep duration (≥9 h vs. 7–8 h). Among the individuals included in the study, 106,192 reported short sleep duration (≤6 h), 34,184 reported long sleep duration (≥9 h), and 305,742 reported a sleep duration of 7–8 h. Participants who reported using any sleep medication and those with extreme self-reported sleep durations (<3 h or >18 h) were excluded from the study.

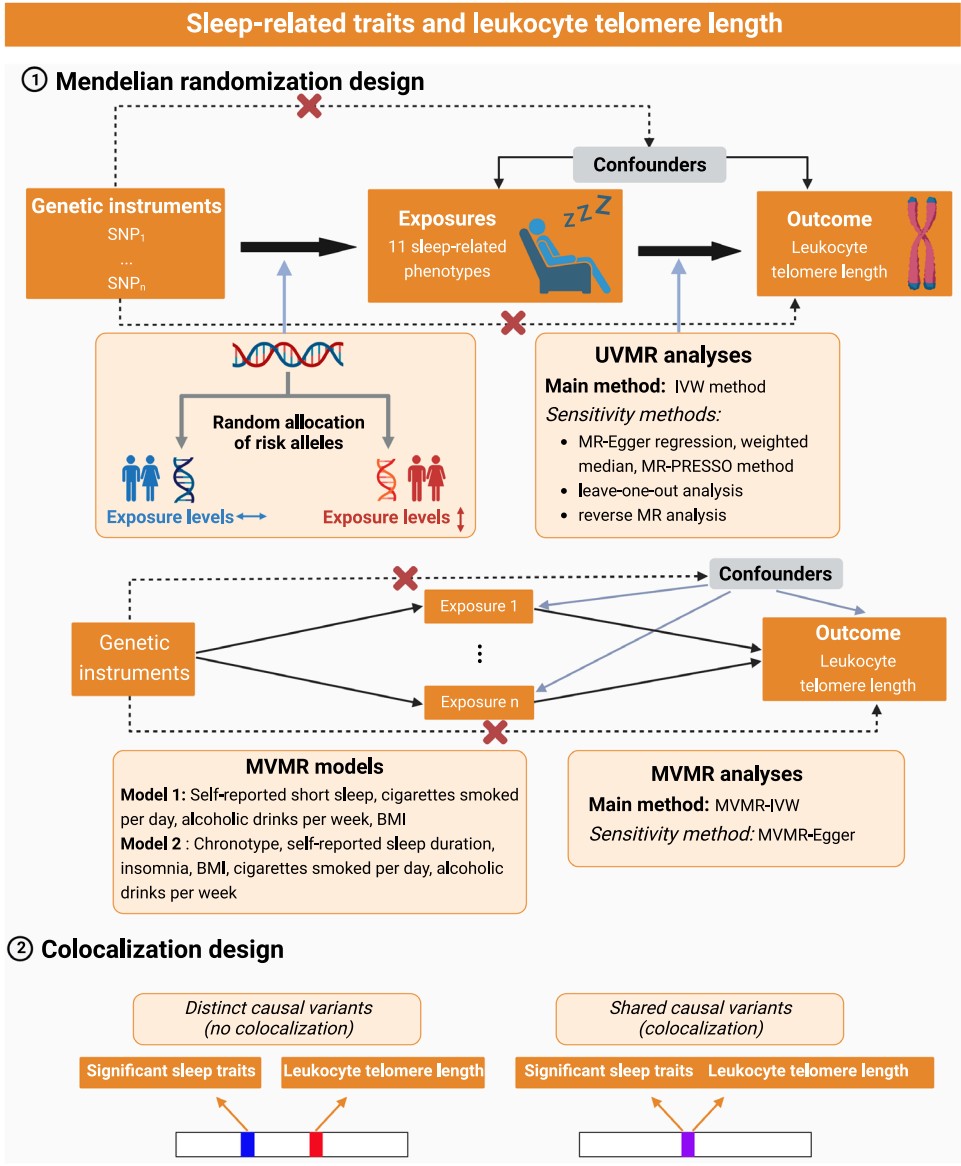

**Fig. 5 Assumptions and study design of the MR study of the associations between sleep-related traits and leukocyte telomere length.** Abbreviations: IVW inverse-variance weighted, MR-PRESSO Mendelian randomization pleiotropy residual sum and outliers, MVMR multivariable Mendelian randomization, SNP single nucleotide polymorphism, UVMR univariable Mendelian randomization. This picture was created with https://www.biorender.com/.

**Table 1 Data sources used in the MR analyses for the current study.**

| Phenotype | Participants included in analysis | Ancestry | Measurement | Adjustments | Source |
|---|---|---|---|---|---|
| **Exposures** | | | | | |
| Self-reported sleep duration | 446,118 individuals | European | Self-reported | Age, sex, 10 PCs, and genotyping array | UK Biobank[37] |
| Self-reported short duration[a] | 106,192 cases (≤6 h) vs. 305,742 controls | European | Self-reported | Age, sex, 10 PCs, and genotyping array | UK Biobank[37] |
| Self-reported long duration[a] | 34,184 cases (≥9 h) vs. 305,742 controls | European | Self-reported | Age, sex, 10 PCs, and genotyping array | UK Biobank[37] |
| Insomnia | 129,270 cases (frequent insomnia) vs. 108,357 controls | European | Self-reported | age, sex, 10PCs, and genotyping array | UK Biobank[62] |
| Chronotype | 252,287 cases (morning chronotype) vs. 150,908 controls | European | Self-reported | Age, sex, study center, and genotyping array | UK Biobank[38] |
| Daytime sleepiness | 452,071 individuals | European | Self-reported | Age, sex, 10 PCs, genotyping array, and genetic correlation matrix | UK Biobank[40] |
| Daytime napping | 452,633 individuals | European | Self-reported | Age, sex, 10 PCs, genotyping array, and genetic correlation matrix | UK Biobank[41] |
| Accelerometer-based sleep duration[b] | 84,810 individuals | European | Accelerometer | Age at accelerometry, sex, study center, season when activity monitor worn, and genotyping array | UK Biobank[42] |
| L5 timing[b] | 85,205 individuals | European | Accelerometer | Age at accelerometry, sex, study center, season when activity-monitor worn, and genotyping array | UK Biobank[42] |
| Sleep efficiency[b] | 84,810 individuals | European | Accelerometer | Age at accelerometry, sex, study center, season when activity-monitor worn, and genotyping array | UK Biobank[42] |
| Number of sleep episodes[b] | 84,441 individuals | European | Accelerometer | Age at accelerometry, sex, study center, season when activity-monitor worn, and genotyping array | UK Biobank[42] |
| **Outcome** | | | | | |
| Leukocyte telomere length | 472,174 individuals | European | multiplex qPCR methodology (T/S ratio) | Age, sex, genotyping array, and 10 PCs | UK Biobank UK Biobank[43] |
| **Confounders** | | | | | |
| BMI | 681,275 individuals | European | Height and weight measured at baseline | Age, sex, recruitment center, genotyping batches, and 10 PCs | GIANT consortium and UK Biobank[46] |
| Alcohol drinks per week | 666,978 individuals | European | Self-reported | Age, sex, age × sex interaction, and 10 genetic PCs | GSCAN consortium Phase 2[45] |
| Cigarettes smoked per day | 326,497 individuals | European | Self-reported | Age, sex, age × sex interaction, and 10 genetic PCs | GSCAN consortium Phase 2[45] |

Abbreviations: GIANT genetic investigation of anthropometric traits, GSCAN GWAS and sequencing consortium of alcohol and nicotine use, L5 timing least active 5 h timing, PCs principal components.
[a]The reference group was participants reported a sleep duration of ≥7 h and <9 h every 24 h (including naps).
[b]Accelerometer measures the activity of the participant's sleep episodes were defined as periods of at least 5 min with no change larger than 5° associated with the z-axis (movement in the dorsal-ventral direction) of the activity-monitor (number of sleep episodes ≤5 or ≥30 were excluded). The summed duration of all sleep episodes was used as the indicator of sleep duration (sleep duration <3 h or >12 h was excluded). Sleep efficiency was calculated as sleep duration divided by the time of the sleep period time window. The least active 5 h timing was estimated using a rolling 5 h time window starting from the previous midnight. The midpoint of the least active 5 h was used in the analysis.

Insomnia was assessed by self-reported responses to a question that inquired about difficulties falling asleep at night or waking up in the middle of the night. Participants who responded "usually" were classified as having frequent insomnia ($N = 129{,}270$), whereas those who responded "never/rarely" were considered as controls ($N = 237{,}627$).

Chronotype refers to an individual's natural inclination toward being a morning person, an evening person, or somewhere in between[38]. Morning chronotype was assessed in the UK Biobank using self-reported measurement by answering the question of whether they were morning or evening people. Participants who reported "Definitely an 'evening' person" and "More an 'evening' than 'morning' person" were classified as controls ($N = 150{,}908$), and those who reported "Definitely a 'morning' person" and "More a 'morning' than 'evening' person" were classified as cases ($N = 252{,}287$). BOLT-LMM with adjustment for age, sex, study center, and genotyping array was performed to estimate the genetic associations between SNPs and morning chronotype[38]. To transform the linear scale of BOLT-LMM association statistics into log odds ratio for the genetic associations with morning chronotype, we utilized the approximation $logOR \approx \beta/(\mu(1-\mu))$, where $\beta$ represents the reported effect size from the BOLT-LMM and $\mu$ is the case fraction of the binary trait ($\mu = 62.6\%$)[39].

Summary statistics for daytime sleepiness ($N = 452{,}071$) and napping ($N = 452{,}633$) were both obtained from the UK Biobank[40,41]. Daytime sleepiness and napping were ascertained by asking the questions "How likely are you to dose off or fall asleep during the daytime when you don't mean to? (e.g., when working, reading, or driving)" and "Do you have a nap during the day?", respectively.

Summary-level data for accelerometer-based sleep traits were obtained from ~85,000 individuals of European ancestry from the UK Biobank ()[42]. Our analysis was centered on 4 sleep traits measured via accelerometers, referring to L5 timing, number of sleep episodes, sleep duration, and sleep efficiency. One Sleep episode within the sleep period time window (SPT-window) was defined as periods of at least 5 min with a change <5° associated with the z-axis. The summed duration of all sleep episodes was calculated to define accelerometer-based sleep duration ($N = 84{,}810$). L5 timing ($N = 85{,}205$) was defined as the midpoint of the 5 h period with the minimum average acceleration of each day. The number of sleep episodes ($N = 84{,}441$) was defined as the number of sleep episodes within the SPT window. Sleep efficiency ($N = 84{,}810$) was calculated as sleep duration divided by the duration of the SPT window.

**Outcome**. The present study utilized the summary statistics for LTL, which were derived from a GWAS meta-analysis conducted on a large sample of 472,174 individuals of European ancestry in the UK Biobank[43]. LTL was quantified using the multiplex quantitative polymerase chain reaction (qPCR) methodology, which measures the ratio of telomere repeat copy number (T) relative to a single copy gene, providing a reliable and precise assessment of LTL[44]. In this process, LTL measurements were log-transformed and Z-standardized to minimize the impact of measurement variability on the results.

**Potential confounders**. MVMR analyses were utilized to account for the potential confounding effects of other risk factors on LTL. With regard to additional exposures included in MVMR models, summary statistics of both smoking (cigarettes smoking per day) and drinking (alcohol drinking per week) were obtained from phase 2 of GWAS and sequencing consortium of alcohol and nicotine use (GSCAN Phase 2)[45], while summary statistics of BMI were obtained from UK Biobank and genetic investigation of

anthropometric traits (GIANT) consortium[46]. Detailed information regarding these genetic datasets is displayed in Table 1.

**Genetic instrumental variable selection**. In order to satisfy the assumptions of MR (Fig. 1), we first selected independent (linkage disequilibrium (LD) $r^2 < 0.001$ within 10 Mb) and genome-wide significant ($P < 5 \times 10^{-8}$) SNPs as genetic IVs to proxy sleep-related traits in UVMR and MVMR analyses. The 1000 Genomes European data were used as the reference for LD $r^2$ estimation. Second, if an SNP is not present in the outcome GWAS, we substituted it with a proxy SNP in high LD ($r^2 > 0.80$), and if a suitable proxy SNP was not available, the SNP was discarded. Finally, we quantified the strength of SNPs for UVMR by calculating the mean $F$-statistic[47], whereby the mean $F$-statistic greater than 10 indicated adequate strength and ensured the validity of the SNPs for the exposures in UVMR.

**Statistics and reproducibility**. After data extraction and harmonization, we performed UVMR analyses of 11 sleep-related traits on LTL. The inverse variance weighting (IVW) method was applied to estimate the effect sizes in the primary analysis. In cases where heterogeneity was detected, the random-effects IVW model was applied, while for all other cases, the fixed-effects IVW model was used[48]. Heterogeneity in the IVW estimates was examined by the Cochran $Q$ test and $I^2$ index.

For significant sleep traits that were detected in UVMR analysis, we leveraged the MVMR-IVW method to identify whether the effects of genetic liability to these sleep traits on LTL are independent of smoking, alcohol consumption, BMI, and other sleep traits[49]. Heterogeneity in the MVMR analysis was examined by the Cochran $Q$ test. For MVMR, conditional $F$-statistics were calculated for the exposures, and larger than 10 indicates a lower risk of weak instrument bias[50].

In the current study, we leveraged approximate Bayes factor (ABF) localization analysis to evaluate whether significant sleep traits and LTL share a common genetic causal variant in the gene region[51]. Colocalization analysis was performed by generating ±200 kb windows from the SNPs used to instrument these sleep traits. ABF computation with default parameters was carried out to quantify the posterior probability that the identified sleep trait and LTL share the same causal signal (Hypothesis 4: both traits share a causal variant in the gene region)[51]. As a convention, posterior probability for H4 (PP.H4) ≥ 80% was considered as evidence for colocalization, indicating a shared genetic variant between significant sleep traits and LTL and a common genetic mechanism may be involved in regulating the association[52].

Firstly, we evaluated the use of other MR methods in UVMR, including MR-Egger regression[53], weighted median[54], and MR-PRESSO[55]. Additionally, we performed radial MR-IVW and radial MR-Egger analyses using modified second-order weights to identify outliers and repeated the UVMR analyses without these outliers as sensitivity analyses[56]. Briefly, radial MR-IVW and radial MR-Egger analyses identify outliers that make large contributions to Cochran's $Q$ statistic or Rucker's $Q$ statistic for heterogeneity, respectively. Second, MR-Steiger filtering was performed to remove reverse causal genetic IVs, which are SNPs that explain more variance in the outcome than in the exposure[57]. After removing reverse causal SNPs, we repeated the MR analyses. Third, for the purpose of identifying outlying IVs across the primary analyses, a leave-one-out analysis was performed where we excluded one SNP at a time and conducted IVW on the remaining SNPs. Fourth, we removed pleiotropic SNPs of 11 sleep-related traits to avoid horizontal pleiotropy and repeated the UVMR analyses of 11 sleep-related traits on LTL. The associations between genetic IVs and various potential confounders, including physical activity, BMI, Townsend deprivation index,

smoking, alcohol consumption, and waist-to-hip ratio, were identified using the PheWAS platform[58]. Any SNPs at a $P$-value of $1 \times 10^{-5}$ were removed in the sensitivity analysis. Fifth, we conducted sensitivity analyses of MVMR that are robust to pleiotropy, including MVMR-Egger, MVMR-PRESSO, and MVMR-least absolute shrinkage and selection operator (MVMR-LASSO)[59]. Furthermore, we conducted the MVMR-Q(het) method, which aimed to minimize the $Q$-statistics and allow for heterogeneity[50]. Sixth, we leveraged the reverse MR method to assess whether LTL showed evidence of causally impacting significant sleep-related traits rather than vice versa. Finally, we employed multiple methods to assess the potential impact of bias introduced by partial sample overlap between sleep traits and LTL on our findings. Specifically, (i) we evaluated the bias and type 1 error rate for sample overlap using an online calculator[60], and (ii) we utilized a robust MR method, MRlap, designed to mitigate bias introduced by sample overlap, winner's curse, and weak instruments[61].

In the UVMR analysis, we employed the false discovery rate (FDR) method to correct for multiple testing, setting the significance threshold at FDR-corrected $P$-value < 0.05. For all other analyses, we considered results with $P$-values < 0.05 as indicative of statistically significant associations. All analyses were conducted using various R software (version 4.2.2) packages, including TwoSampleMR (version 0.5.6), MVMR (version 0.3), coloc (version 5.1.0.1), locuscomparer (version 1.0.0), MR-PRESSO (version 1.0), MRlap (version 0.0.3.0), MendelianRandomization (version 0.7.0), and RadialMR (version 1.1), with a two-sided approach.

**Reporting summary**. Further information on research design is available in the Nature Portfolio Reporting Summary linked to this article.

## Data availability

Only publicly available data were used in this study, and data sources and handling of these data are described in the Methods and Supplementary Data 1–13. Summary-level data on sleep-related traits could be obtained from https://sleep.hugeamp.org/downloads.html, and summary-level data on telomere length could be obtained from https://figshare.com/s/caa99dc0f76d62990195. The source data used to plot Fig. 1 are in Supplementary Data 3. The source data used to plot Fig. 2 are in Supplementary Data 1. The source data used to plot Fig. 3 are in Supplementary Data 9 and Data 10. Further information is available from the corresponding author upon request.

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

## Acknowledgements

This study was supported by the National Natural Science Foundation of China (82200933), the Natural Science Foundation of Hunan Province for Youths (2022JJ40689), and the Natural Science Foundation of Changsha (kq2202404). We acknowledge the funding for their support. We greatly appreciate the GIANT consortium for providing summary-level data on BMI and the GSCAN consortium for providing summary-level data on smoking and drinking. We also greatly appreciate the authors of the cited data source for providing summary statistics of sleep-related traits and telomere length.

## Author contributions

All the authors contributed to the paper review and editing. They all approved to submission of the final version of the paper. J.Y.H., J.W.L., Z.Q.W., and Z.X.G. contributed to the research questions and study design. J.Y.H., J.W.L., and Z.Q.W. contributed to the data curation. J.Y.H., J.W.L., Q.H.L., and Z.Q.W. contributed to the methodology development. Z.Q.W., J.Y.H., J.W.L., and Q.H.L. conducted statistical analyses. Z.Q.W., J.W.L., J.Y.H., and W.P.W. helped validate and perform sensitivity analyses. J.Y.H., Z.Q.W., J.W.L., Q.H.L., and W.P.W. interpreted the results and wrote the original draft of the paper. Z.Q.W., J.Y.H., J.W.L., Q.H.L., W.P.W., and Z.X.G. helped review and edit the final draft of the paper. The author read and approved the final paper.

## Competing interests

The authors declare no competing interests.
