## [Peer Review File · Communications Biology]

reviewers' comments:

Reviewer #1 (Remarks to the Author):

The authors sought to understand the conflicting observational literature on sleep and LTL by using MR approaches. They suggest short sleep duration and evening chronotype are causal determinants of short LTL. They found evidence for genetic pleiotropy between sleep and LTL using colocalization.

Strengths of the work are use of accelerometer derived sleep variables (self-report are notoriously unreliable), use of MR robust methods, reverse and MVMR, and colocalization.

I have only a few suggestions for additional analyses which would in my view strengthen the paper:

1. There are potential biases in one sample MR which the authors comment on. MRlap is a new method to try and adjust for overlapping samples which might be worth doing.
2. What was the rationale for binarising self-reported sleep into short and long duration? Is it not more powerful to keep as total sleep? If non-linear relationships are of interest non-linear MR could be employed.
3. GSCAN sum stats are used for alcohol and smoking in MVMR. GSCAN 2 has now been released with larger numbers (Saunders et al., 2022, Nature).
4. Please include MR scatterplots.
5. P value thresholds - might be better to use FDR thresholds (in addition to Bonferroni) rather than talking about suggestive significance.

Reviewer #2 (Remarks to the Author):

The authors investigated the association between sleep traits (both derived from questionnaire and accelerometer) and leukocyte telomere length using Mendelian Randomization. This is overall a well-conducted and written study. I have a number of comments to be addressed by the authors:

- Note that Mendelian Randomization is only a tool to provide evidence for possible causality. It is not a way to prove causality. sentences like "insight into significant causations" are therefore way too strong and technically incorrect. Please rephrase this throughout the manuscript.
- The two samples (sleep + LTL) were both derived from the UK Biobank, and therefore contain largely overlapping samples which can result into bias (Winner's curse), and is therefore a serious potential concern. The study would benefit if they can include an independent sample to validate their results (either sleep traits or the LTL derived from another study). The authors provide some data into the possible bias, but this is rather unclear for a general reader (sentences 272-275 are unclear and need clarification).
- Added value of the colocalization analysis in the context of the study aim is unclear to this reviewer.
- Multiple testing correction might be too stringent given the partly overlap between the phenotypes?
- The effect sizes with chronotype are really really small; although the authors corrected for this, I wonder whether this went correct? Standard errors are also way smaller than with other sleep traits (with a similar number of IVs)
- I appreciate the reverse MR; did the authors also consider MR Steiger filtering?
- Figure 3: not sure what is presented here. Why not presenting leave-one out plots?

-

Reviewer #3 (Remarks to the Author):

The study related eleven sleep traits to leukocyte telomere length (LTL) using univariable and

multivariable MR. It found a strong causal relationship between short sleep and reduced LTL, while a suggestive causal effect between morning chronotype and longer LTL.

Here, I have some comments.

1. 'In this study, we aimed to comprehensively investigate the causal associations between sleep traits and LTL by conducting two-sample MR analyses on a set of 7 self-reported and 4 accelerometer-based sleep traits.' Self-report bias might exist for some variables. The authors should consider it.

2. As to the data, all exposure and outcome are from summary statistics online. The study has a detailed description of the definition and processing of variables. However, this could be placed in the supplementary file and replaced with a table.

3. The problem of most concern is the overlap of the sample. Even though the instruments are strong, bias relies on the covariance of associations between the variant-exposure and variant-outcome in summarized data. The methods to assess bias from sample overlap is still not convincing, more ways of assessment can be considered (e.g., <https://sb452.shinyapps.io/overlap/>). However, why did the authors use the definitely UKB samples as an outcome, rather than another, e.g., FinnGen?

4. In using MVMR, did the authors consider using some strategies like in UVMR (Cochran Q test and I2 index) to exclude outliers and check heterogeneity? For F statistics, conditional F-statistic can be presented for MVMR.

Reviewers' Comments to Author:

Reviewer #1

COMMENTS TO AUTHOR(S)

The authors sought to understand the conflicting observational literature on sleep and LTL by using MR approaches. They suggest short sleep duration and evening chronotype are causal determinants of short LTL. They found evidence for genetic pleiotropy between sleep and LTL using colocalization.

Strengths of the work are use of accelerometer derived sleep variables (self-report are notoriously unreliable), use of MR robust methods, reverse and MVMR, and colocalization.

I have only a few suggestions for additional analyses which would in my view strengthen the paper:

Response: We sincerely appreciate the time and effort that you dedicated to providing positive feedback and insightful comments on our manuscript. We have addressed all the comments point by point and made changes and improvements in our revised manuscript accordingly. Specifically, we have i) performed MRlap analysis to consider the impact of bias introduced by sample overlap on our findings; ii) performed MVMR analysis using updated summary statistics of smoking and drinking (GSCAN 2); iii) complemented scatter plots; and iv) utilized the FDR-corrected method instead of the Bonferroni method to account for the correction of multiple testing. The results were not altered. All changes in the manuscript are tracked in the marked-up version and the line numbers in the following responses refer to the “marked-up” version of the manuscript.

1. There are potential biases in one sample MR which the authors comment on. MRlap is a new method to try and adjust for overlapping samples which might be worth doing.

Response: Thank you very much for your comment. We have considered this issue and previously conducted a sensitivity analysis to evaluate the influence of bias introduced by sample overlap on the results (lines 394-397, page 18). The comments you provided have prompted us to consider the issue more comprehensively. To comprehensively investigate whether the degree of bias introduced by sample overlap impacted our findings, we performed MRlap analysis to assess the relationships between sleep-related traits and LTL. MRlap is a recently developed MR method which can account and correct for bias arising from sample overlap, weak instruments and winner's curse¹. It leveraged cross-trait linkage disequilibrium score regression (LDSC) intercept to approximate the impact of sample overlap bias¹. To implement the MRlap method, we utilized the “MRlap()” function from the “MRlap” package (version 0.0.3.0). MRlap was performed using the all GWAS summary statistics of sleep-related traits and LTL. MRlap estimates were largely similar to the primary analysis (**Table R1**). Specifically, genetically predicted self-reported short sleep was associated with decreased LTL (Corrected β [SE]: -0.123 [0.062]; $P=0.047$), while morning chronotype was associated with increased LTL (Corrected β [SE]: 0.024 [0.012]; $P=0.036$; **Table R1**).

We have also provided the results in *Supplementary Table S12* and incorporated MRlap

analysis to the main text, as follow:

1. lines 292-297, page 14: “Finally, we employed multiple methods to assess the potential impact of bias introduced by partial sample overlap between sleep traits and LTL on our findings. Specifically, i) we evaluated the bias and type 1 error rate for sample overlap using an online calculator (<https://sb452.shinyapps.io/overlap/>)², and ii) we utilized a robust MR method, MRlap, designed to mitigate bias introduced by sample overlap, winner’s curse and weak instruments”

2. lines 308-311, page 14: “All analyses were conducted using various R software (version 4.2.2) packages including TwoSampleMR (version 0.5.6), MVMR (version 0.3), coloc (version 5.1.0.1), locuscomparer (version 1.0.0), MR-PRESSO (version 1.0), and MRlap (version 0.0.3.0), with a two-sided approach.”

3. lines 400-401, page 18: “Additionally, the MRlap analyses showed concordant results with primary results (Supplementary Table S12).”

Additionally, we assessed the bias and type 1 error rate for sample overlap using an online calculator (<https://sb452.shinyapps.io/overlap/>)². The type 1 error rate due to sample overlap between 11 sleep-related traits and LTL was controlled under 0.05 and the biases were minimal (Supplementary Table S11). Hence, despite the considerable overlap between the two samples, considerable weak instrument bias would not be expected. Additionally, we have discussed the bias in the limitation (lines 521-530, page 23).

“First, sample overlap of the current two-sample MR study due to both samples of sleep-related traits and LTL being from UK Biobank may lead to bias and type 1 error (false positive) inflation. For this, the type 1 error rate due to sample overlap between sleep-related traits and LTL was controlled under 0.05, and the MR estimates were well validated by using the MRlap method. Furthermore, a recent simulation study provided support for the rationality and validity of two-sample MR using overlapping samples in large cohorts³.”

Table R1. MRlap estimates for the causal associations between sleep-related traits and leukocyte telomere length.

Exposure	Outcome	Observed β	Observed SE	Observed P	Corrected β	Corrected SE	Corrected P	Test difference	P for difference
Chronotype	LTL	0.018	0.009	0.034	0.024	0.012	0.036	-1.154	0.248
Insomnia	LTL	-0.011	0.033	0.735	-0.014	0.048	0.766	0.232	0.817
Self-reported sleep duration	LTL	0.069	0.047	0.144	0.094	0.064	0.141	-1.513	0.13
Self-reported short sleep	LTL	-0.089	0.045	0.045	-0.123	0.062	0.047	1.878	0.081
Self-reported long sleep	LTL	0.064	0.054	0.239	0.097	0.078	0.215	-1.471	0.141
Daytime napping	LTL	-0.025	0.027	0.361	-0.026	0.034	0.439	0.215	0.829
Daytime sleepiness	LTL	0.012	0.047	0.787	0.025	0.066	0.701	-0.694	0.488
Accelerometer-based sleep duration	LTL	-0.011	0.024	0.638	-0.014	0.034	0.672	0.303	0.762
L5 timing	LTL	-0.007	0.025	0.772	-0.009	0.036	0.781	0.246	0.806
Sleep efficiency	LTL	-0.019	0.061	0.752	-0.029	0.093	0.757	0.297	0.766
No. of sleep episodes	LTL	-0.002	0.031	0.955	-0.002	0.043	0.961	0.031	0.976
LTL	Chronotype	0.002	0.013	0.874	0.003	0.014	0.854	-0.279	0.780
LTL	Self-reported short sleep	-0.013	0.011	0.249	-0.013	0.012	0.271	0.118	0.906

Abbreviations: LTL, leukocyte telomere length; SE, standard error.
All statistical tests were two-sided. $P < 0.05$ was considered significant.

2. What was the rationale for binarising self-reported sleep into short and long duration? Is it not more powerful to keep as total sleep? If non-linear relationships are of interest non-linear MR could be employed.

Response: Thank for your valuable comment. First, the binarization process of short and long sleep has been performed in the original GWAS data of the UK Biobank.⁴ “Sleep duration was treated as a continuous variable and also categorized as either short (6 h or less), normal (7 or 8 h), or long (9 h or more) sleep duration.” This approach is rooted in the premise put forth by the referenced study, indicating that both short (≤ 6 h per night) and long (≥ 9 h per night) habitual self-reported sleep durations are associated with various cognitive, psychiatric, metabolic, cardiovascular, immunological dysfunctions, and all-cause mortality, as compared to the reference sleep duration of 7-8 h per night⁵⁻⁷.

Secondly, while using total sleep duration could capture more nuanced variations in sleep patterns, the binarization approach allowed us to explore potential threshold effects or non-linear relationships between sleep duration and the outcome. This approach is useful in two-sample MR analysis to investigate whether there exists a critical point of sleep duration beyond which its impact on the outcome becomes more pronounced or diminishes⁸.

Thirdly, we fully agree with the suggestion of employing non-linear MR to investigate dose-response relationships and other non-linear associations between sleep duration and the outcome. But it is essential to consider the challenges associated with obtaining individual-level data. This aspect has been duly discussed in the limitations section of our article and we encourage larger research projects to confirm or further develop the findings of our study.

3. GSCAN sum stats are used for alcohol and smoking in MVMR. GSCAN 2 has now been released with larger numbers (Saunders et al., 2022, Nature).

Response: Thank you very much for your comment. We have performed the MVMR analysis in the main text using the updated summary statistics of alcohol consumption and smoking from Phase 2 GSCAN consortium⁹. According to publicly available data their provided (<https://conservancy.umn.edu/handle/11299/241912>), summary statistics of alcoholic drinks per week were obtained from a meta-analysis of GWASs within a total of 666,978 individuals of European ancestry and summary statistics of cigarettes smoked per day were derived from a meta-analysis of GWASs within a total of 326,497 individuals of European ancestry. The updated findings are presented in Figure 4 within the main text, and for ease of reading, they are also included in **Figure R1** in the letter. The updated results were similar with previous results. Using the updated summary statistics of drinking and smoking, MVMR showed the inverse direct effect of self-reported short sleep duration on LTL (β [95% CI]: -0.159 [-0.310, -0.009]; $P=0.038$) after adjusting for the effect of BMI, smoking and alcohol (**Figure R1**). After adjusting for the effect of self-reported sleep duration, insomnia, BMI, smoking, and alcohol consumption, we observed that MVMR results supported the findings of a positive effect of morning chronotype on LTL (0.017 [0.001, 0.033]; $P=0.048$; **Figure R1**). MVMR-Egger provided concordant results and indicated no evidence of horizontal pleiotropy (*Supplementary Table S8 and S9*). We have used the updated summary data of smoking and drinking and revised the corresponding results in the main text, as

follow:

1. lines 232-235, page 11: “With regard to additional exposures included in MVMR models, summary statistics of both smoking (cigarettes smoking per day) and drinking (alcohol drinking per week) were obtained from phase 2 of GWAS and sequencing consortium of alcohol and nicotine use (GSCAN Phase 2)⁹.”

2. lines 372-378, page 17: “MVMR-IVW method showed the inverse direct effect of self-reported short sleep duration on LTL (-0.159 [-0.310, -0.009]; P=0.038) after adjusting for the effect of BMI, smoking, and alcohol consumption (Figure 4). After adjusting for the effect of self-reported sleep duration, insomnia, BMI, smoking, and alcohol consumption, we observed that MVMR results supported the findings of a positive effect of morning chronotype on LTL (0.017 [0.001, 0.033]; P=0.048; Figure 4).”

Additionally, we also used GSCAN2 summary statistics of smoking and drinking to exclude potential pleiotropic SNPs in the sensitivity analysis. However, compared with GSCAN phase 1, no additional SNPs were excluded.

Figure R1. The direct effect of genetically determined self-reported short sleep duration and morning chronotype on LTL using MVMR adjusted for smoking, alcohol consumption, BMI and other sleep traits.

Abbreviations: BMI, body mass index; LTL, leukocyte telomere length; SNPs, single-nucleotide polymorphisms.

The reference group of self-reported short sleep (≤ 6 hours) is sleep duration between ≥ 7 hours and < 9 hours in every 24 hours. The reference group of chronotype is evening preference chronotype.

4. Please include MR scatterplots.

Response: We appreciate you a lot for your suggestion. In the manuscript, we have provided scatter plots in the Supplementary materials (*Supplementary Figure S1*) and incorporated MR scatter plots illustrating the genetic association between 11 sleep-related traits and LTL, as follow (lines 334-336, page 15):

“Scatter plots depicting the genetic associations with LTL against the genetic associations with the sleep-related traits were provided (Supplementary Figure S1).”

We also provided the MR scatterplots in the letter (**Figure R2**).

Figure R2. Scatter plot of genetic association with LTL against associations with A) self-reported sleep duration, B) self-reported short sleep, C) self-reported long sleep, D) insomnia, E) morning chronotype, F) L5 timing, G) No. of sleep episodes, H) sleep efficiency, I) accelerometer-based sleep duration, J) daytime sleepiness, and K) daytime napping.

Abbreviations: L5 timing, least active 5 hours timing; LTL, leukocyte telomere length; SD, standard derivation.

The associations (β -coefficients) of A) self-reported sleep duration, B) self-reported short sleep, C) self-reported long sleep, D) insomnia, E) morning chronotype, F) L5 timing, G) No. of sleep episodes, H) sleep efficiency, I) accelerometer-based sleep duration, J) daytime sleepiness, and K) daytime napping on LTL were indicated by red solid lines using the inverse-variance weighted (IVW) method. Scatters showed associations of SNPs with sleep-related traits and LTL, with dotted cross-hairs indicating standard errors (SE).

5. *P* value thresholds - might be better to use FDR thresholds (in addition to Bonferroni) rather than talking about suggestive significance.

Response: Thank you very much for your comment. We agree with your perspective. Instead of Bonferroni method, we employed the FDR approach (also known as the Benjamini-Hochberg method) to address the multiple testing concern in the UVMR analysis. We set the significance threshold at FDR-corrected P -value <0.05 . We have made modifications to **Figure 2** in order to display the FDR-corrected P -values of the results in the main text. Additionally, we have included the updated Figure 2 as **Figure R3** in the letter for reference. After using FDR-corrected method, we observed genetically predicted self-reported short sleep duration was associated with decreased LTL (β [95% CI]: -0.315 [-0.451, -0.178]; FDR-corrected $P<0.001$). Genetically predicted morning chronotype was associated with increased LTL (0.016 [0.004, 0.028]; FDR-corrected $P=0.049$; **Figure R3**). Moreover, we have revised the methods and results description in the manuscript, as follow:

1. lines 63-65, page 3: “*UVMR indicated that genetically determined short sleep was associated with decreased LTL, while morning chronotype was associated with increased LTL.*”
2. lines 302-304, page 14: “*In the UVMR analysis, we employed the false discovery rate (FDR) method to correct for multiple testing, setting the significance threshold at FDR-corrected P -value <0.05 .*”
3. lines 320-322, page 15: “*One-unit higher log odds of self-reported short sleep duration decreased 0.315 standard derivation (SD) of LTL (β [95% CI]: -0.315 [-0.451, -0.178]; FDR-corrected $P<0.001$; Figure 2).*”
4. lines 324-327, page 15: “*The IVW method showed genetically predicted morning chronotype was associated with the longer LTL (0.016 [0.004, 0.028]; FDR-corrected $P=0.049$), compared with the evening chronotype (Figure 2).*”

Figure R3. The effect of genetically determined sleep-related traits on LTL using UVMR.

Abbreviations: LTL, leukocyte telomere length; SNPs, single-nucleotide polymorphisms; L5 timing, least active 5 hours timing.

Reviewer #2

COMMENTS TO AUTHOR(S)

The authors investigated the association between sleep traits (both derived from questionnaire and accelerometer) and leukocyte telomere length using Mendelian Randomization. This is overall a well-conducted and written study. I have a number of comments to be addressed by the authors:

Response: Thank you for your positive assessment of our study and for recognizing its comprehensiveness and methodological soundness. We have examined our manuscript very carefully and addressed all the revisions point by point as follows. Specifically, we have i) performed MRlap analysis and estimate type 1 error rate and bias to consider the impact of bias introduced by sample overlap on our findings; ii) complemented leave-one-out plot in the Supplementary material; iii) utilized the FDR-corrected method instead of the Bonferroni method to account for the correction of multiple testing; and iv) meticulously checked SNP effect sizes for chronotype and rectified the estimation of morning chronotype on LTL. All changes in the manuscript are tracked in the marked-up version and the line numbers in the following responses refer to the “marked-up” version of the manuscript. We hope that these revisions demonstrate our commitment to addressing your concerns and that they make the study more suitable for publication.

- Note that Mendelian Randomization is only a tool to provide evidence for possible causality. It is not a way to proof causality. sentences like "insight into significant causations" are therefore way too strong and technically incorrect. Please rephrase this throughout the manuscript.

Response: Thank you for your valuable comments. We have thoroughly revised the manuscript, ensured precise and accurate language while maintained a balanced and objective tone. We have taken care to remove any instances of an overly critical or affirmative tone, aiming to enhance the manuscript's overall readability and clarity. We acknowledge the limitations of MR and will employ appropriate terminology to convey its evidence-based nature in inferring causal associations. Your feedback has been instrumental in improving the manuscript, and we sincerely appreciate your valuable contributions.

For example:

lines 55-59, page 3: *“The study aimed to determine the causal associations between 11 sleep-related traits and leukocyte TL (LTL) through two-sample Mendelian randomization (MR) and colocalization analyses using the summary statistics from large-scale genome-wide association studies.”*

lines 108-116, page 6: *“Moreover, given the nature of observational studies, they may not be sufficiently robust in establishing cause-and-effect relationships. To address this issue, mendelian randomization (MR) studies are commonly used to infer causality between an exposure and an outcome using genetic variants randomly allocated at conception¹⁰. Nonetheless, the only MR study reported conflicting results, showing no causal relationship between insomnia and LTL¹¹. Thus, additional evidence is imperative to disentangle the correlation between sleep and LTL, with particular emphasis on different types of sleep patterns.”*

lines 251-256, page 12: *“After data extraction and harmonization, the effect size of 11 sleep-related traits on LTL was estimated using the inverse variance weighting (IVW) method in the primary analysis.”*

lines 409-411, page 19: *“Reverse MR indicated that genetically predicted LTL does not*

exert an effect on chronotype and short sleep duration.”

- The two samples (sleep + LTL) were both derived from the UK Biobank, and therefore contain largely overlapping samples which can result into bias (Winner's curse), and is therefore a serious potential concern. TH study would benefit if they can include an independent sample to validate their results (either sleep traits or the LTL derived from another study). The authors provide some data into the possible bias, but this is rather unclear for a general reader (sentences 272-275 are unclear and need clarification).

Response: We appreciate you a lot for your suggestion. We understand the importance of utilizing such non-overlapped exposure and outcome GWAS datasets to avoid potential bias. However, after a comprehensive search in PubMed, web of science, and GWAS catalogue, we have encountered challenges in finding suitable non-overlapping GWAS datasets that align with our research focus. For sleep-related traits, although we need the genome-wide significantly SNPs, genome-wide significantly SNPs were identified using UK Biobank cohort. For LTL, we need the whole summary-level genetic data of LTL, the currently publicly available dataset was analysed using UK Biobank cohort. Given the current limitations in finding non-overlapping datasets, we conducted a series of sensitivity analyses to investigate whether the bias introduced by sample overlap impacted our findings.

For the previous paper (lines 272-275 for previous paper), we estimate the bias via calculating the proportion of overlap in the two samples and the lower limit of a one-sided 95% CI for the F -statistics². For this method, Burgess *et al.* have established that the extent of bias caused by sample overlap depends on both the population F -statistic and the degree of overlap between the studies through extensive simulation studies². For example, if the F -statistic >10 and the percentage of sample overlap is 50%, then the bias is less than 5%. Now, we have improved the sensitivity analysis to more comprehensively assess the impact of bias introduced by sample overlap on our findings. First, we further evaluated the detailed bias and type 1 error rate for sample overlap using an online calculator (<https://sb452.shinyapps.io/overlap/>)². Subsequently, we employed a recently developed MR technique known as MRlap to explore the associations between sleep-related traits and LTL¹. MRlap could account and correct for bias arising from sample overlap, weak instruments and winner's curse. MRlap using cross-trait LDSC to approximate the sample overlap and combats sample overlap, for example, by reducing an observed 15% overestimation for fully overlapping samples to a 5% overestimation¹.

The type 1 error rate due to the sample overlap between sleep-related traits and LTL was controlled under 0.05 and the biases were minimal (*Supplementary Table S11 and Table R2*). MRlap estimates were largely similar to the primary analysis (*Supplementary Table S12 and Table R3*). Specifically, genetically predicted self-reported short sleep was associated with decreased LTL (Corrected β [SE]: -0.123 [0.062]; $P=0.047$), while morning chronotype was associated with increased LTL (Corrected β [SE]: 0.024 [0.012]; $P=0.036$; *Supplementary Table S12 and Table R3*).

We have revised the description of the methods and results of these sensitivity analyses in the manuscript, as follow:

1. lines 292-297, page 14: “Finally, we employed multiple methods to assess the potential impact of bias introduced by partial sample overlap between sleep traits and LTL on our findings. Specifically, i) we evaluated the bias and type 1 error rate for sample overlap using an online calculator (<https://sb452.shinyapps.io/overlap/>)², and ii) we utilized a robust MR method, MRlap, designed to mitigate bias introduced by sample overlap, winner’s curse and weak instruments¹.”

2. lines 398-401, page 18: “The type 1 error rate due to the sample overlap between sleep-related traits and LTL was controlled under 0.05 and the biases were minimal (Supplementary Table S11). Additionally, the MRlap analyses showed concordant results with primary results (Supplementary Table S12).”

We have discussed the bias in the limitation (lines 521-530, page 23).

“First, sample overlap of the current two-sample MR study due to both samples of sleep-related traits and LTL being from UK Biobank may lead to bias and type 1 error (false positive) inflation. For this, the type 1 error rate due to sample overlap between sleep-related traits and LTL was controlled under 0.05, and the MR estimates were well validated by using the MRlap method. Furthermore, a recent simulation study provided support for the rationality and validity of two-sample MR using overlapping samples in large cohorts³.”

Table R2. The degree of overlapping between samples of sleep-related traits and samples of leukocyte telomere length.

Exposures	Sample size of exposure GWAS	Outcomes	Sample size of outcome GWAS	Overlap degree	Number of instruments	R^2	Observational estimate	Bias	Type 1 error rate
Chronotype	403,195	LTL	472,174	85.39%	117	1.296%	0.016 ^a	0.0002	0.05
Insomnia	237,627	LTL	472,174	50.33%	39	0.698%	-0.034 ^a	0.0004	0.05
Self-reported sleep duration	446,118	LTL	472,174	94.48%	66	0.604%	0.055 ^a	0.0013	0.05
Self-reported short sleep	411,934	LTL	472,174	87.24%	24	0.219%	-0.315 ^a	-0.0064	0.05
Self-reported long sleep	339,926	LTL	472,174	71.99%	10	0.117%	0.252 ^a	0.0063	0.05
Daytime napping	452,633	LTL	472,174	95.86%	105	1.061%	-0.034 ^a	-0.0012	0.05
Daytime sleepiness	452,071	LTL	472,174	95.74%	37	0.343%	-0.005 ^a	<0.0001	0.05
Accelerometer-based sleep duration	84,810	LTL	472,174	17.96%	9	0.516%	-0.004 ^a	<0.0001	0.05
L5 timing	85,205	LTL	472,174	18.05%	5	0.347%	-0.013 ^a	<0.0001	0.05
Sleep efficiency	84,810	LTL	472,174	17.96%	5	0.267%	-0.02 ^a	-0.0006	0.05
No. of sleep episodes	84,441	LTL	472,174	17.88%	22	0.982%	-0.002 ^a	<0.0001	0.05

Abbreviations: Genome-wide association study (GWAS); leukocyte telomere length (LTL).

^a Since the observational estimates were not available from previous studies, we applied the estimates in our study as the substitute parameter.

Table R3. MRlap estimates for the causal associations between sleep-related traits and leukocyte telomere length.

Exposure	Outcome	Observed β	Observed SE	Observed P	Corrected β	Corrected SE	Corrected P	Test difference	P for difference
Chronotype	LTL	0.018	0.009	0.034	0.024	0.012	0.036	-1.154	0.248
Insomnia	LTL	-0.011	0.033	0.735	-0.014	0.048	0.766	0.232	0.817
Self-reported sleep duration	LTL	0.069	0.047	0.144	0.094	0.064	0.141	-1.513	0.13
Self-reported short sleep	LTL	-0.089	0.045	0.045	-0.123	0.062	0.047	1.878	0.081
Self-reported long sleep	LTL	0.064	0.054	0.239	0.097	0.078	0.215	-1.471	0.141
Daytime napping	LTL	-0.025	0.027	0.361	-0.026	0.034	0.439	0.215	0.829
Daytime sleepiness	LTL	0.012	0.047	0.787	0.025	0.066	0.701	-0.694	0.488
Accelerometer-based sleep duration	LTL	-0.011	0.024	0.638	-0.014	0.034	0.672	0.303	0.762
L5 timing	LTL	-0.007	0.025	0.772	-0.009	0.036	0.781	0.246	0.806
Sleep efficiency	LTL	-0.019	0.061	0.752	-0.029	0.093	0.757	0.297	0.766
No. of sleep episodes	LTL	-0.002	0.031	0.955	-0.002	0.043	0.961	0.031	0.976
LTL	Chronotype	0.002	0.013	0.874	0.003	0.014	0.854	-0.279	0.780
LTL	Self-reported short sleep	-0.013	0.011	0.249	-0.013	0.012	0.271	0.118	0.906

Abbreviations: LTL, leukocyte telomere length; SE, standard error.
All statistical tests were two-sided. $P < 0.05$ was considered significant.

- Added value of the colocalization analysis in the context of the study aim is unclear to this reviewer.

Response: Thank you for your comment. We conducted colocalization analysis to examine the presence of shared common genetic causal variants in genomic regions between short sleep duration and LTL, as well as between chronotype and LTL (lines 265-267, page 13). If a shared causal variant was found for these pairs, this contributes further evidence that the sleep habits (i.e., short sleep duration and morning chronotype) and LTL share a mechanism, and that any identified MR association is unlikely to be attributable to genetic confounding through a variant in linkage disequilibrium. In our study, colocalization analysis identified that genetic variant rs2517827 was shared between short sleep duration and LTL (PP.H4=97.34%; **Figure 5A**), while rs11712056 was shared between chronotype and LTL, supporting the possibility of a shared genetic basis between these traits (PP.H4=83.88%; **Figure 5B**). This indicates the possibility of a shared genetic basis and underlying mechanism among these traits, strengthening the evidence of the relationships of short sleep duration and morning chronotype with LTL. Furthermore, we have discussed the results in the fourth paragraph in the part of “Discussion” (lines 457-468, page 21).

- Multiple testing correction might be too stringent given the partly overlap between the phenotypes?

Response: Thank you very much for your comment. As you and another reviewer suggested, we have employed FDR method to correct for multiple testing in UVMR analysis. We set the significance threshold at FDR-corrected P -value <0.05 . We have made modifications to **Figure 2** in order to display the FDR-corrected P -values of the results in the main text. Additionally, we have included the updated Figure 2 as **Figure R3** in the letter for reference. After using FDR-corrected method, we observed genetically predicted self-reported short sleep duration was associated with decreased LTL (β [95% CI]: -0.315 [-0.451, -0.178]; FDR-corrected $P<0.001$). Genetically predicted morning chronotype was associated with increased LTL (0.016 [0.004, 0.028]; FDR-corrected $P=0.049$; **Figure R3**). Moreover, we have revised the methods and results description in the manuscript, as follow:

1. lines 63-65, page 3: “UVMR indicated that genetically determined short sleep was associated with decreased LTL, while morning chronotype was associated with increased LTL.”

2. lines 302-304, page 14: “In the UVMR analysis, we employed the false discovery rate (FDR) method to correct for multiple testing, setting the significance threshold at FDR-corrected P -value <0.05 .”

3. lines 320-322, page 15: “One-unit higher log odds of self-reported short sleep duration decreased 0.315 standard derivation (SD) of LTL (β [95% CI]: -0.315 [-0.451, -0.178]; FDR-corrected $P<0.001$; Figure 2).”

4. lines 324-327, page 15: “The IVW method showed genetically predicted morning chronotype was associated with the longer LTL (0.016 [0.004, 0.028]; FDR-corrected $P=0.049$), compared with the evening chronotype (Figure 2).”

Figure R3. The effect of genetically determined sleep-related traits on LTL using UVMR.

Abbreviations: LTL, leukocyte telomere length; SNPs, single-nucleotide polymorphisms; L5 timing, least active 5 hours timing.

- *The effect sizes with chronotype are really really small; although the authors corrected for this, I wonder whether this went correct? Standard errors are also way smaller than with other sleep traits (with a similar number of IVs)*

Response: Thank you for raising this point. We appreciate your careful attention to the effect sizes associated with chronotype in our study. In response to your inquiry, we traced back to address the concerns you raised regarding the accuracy of corrections. It came to our attention that the transformation had inadvertently been applied twice in our initial manuscript. This unintentional repetition led to a distortion of the effect sizes. The complete data transformation procedure is meticulously outlined in the data description document as follows: “*The conversion for BETA to LOGOR is: $LOGOR = BETA / (\mu * (1 - \mu))$, where $\mu = \text{case fraction}$. The same conversion applies to the standard error: $LOGOR_SE = SE / (\mu * (1 - \mu))$.*” (The downloaded file is from <https://www.ebi.ac.uk/gwas/studies/GCST007565>). We have taken prompt action to rectify this issue and have now accurately reinstated the effect sizes of SNPs associated with chronotype. We are pleased to confirm that this rectification did not alter the overall study outcomes. The IVW method showed genetically predicted morning chronotype was associated with the longer LTL (0.016 [0.004, 0.028]; FDR-corrected $P=0.049$), compared with the evening chronotype (**Figure R3**). No horizontal pleiotropy was detected using MR-Egger regression (*Supplementary Table S3*).

Although MR-PRESSO method detected horizontal pleiotropy, the result for morning chronotype (0.016 [0.001, 0.031]; $P=0.032$) was similar with the primary analysis (Supplementary Table S3). We have revised the sentence in the manuscript, as follow:
 1.lines 324-327, page 15: “The IVW method showed genetically predicted morning chronotype was associated with the longer LTL (0.016 [0.004, 0.028]; FDR-corrected $P=0.049$), compared with the evening chronotype (Figure 2).”
 2.lines 330-334, page 15: “Although MR-PRESSO method detected horizontal pleiotropy, the results for self-reported short sleep duration (-0.315 [-0.451, -0.178]; $P<0.001$), morning chronotype (0.016 [0.001, 0.031]; $P=0.032$), and other sleep traits were not substantially altered (all $P>0.05$) after outlier-correction (Supplementary Table S3).”

Subsequently, the **Figure 2**, **Figure 3E** have been updated using the corrected effect sizes. **Figure 2** was presented in the letter as **Figure R3**. **Figure 3** was presented in the letter as **Figure R4**. Your engagement really contributes to the refinement of our study, and we are grateful for your valuable input again.

- I appreciate the reverse MR; did the authors also consider MR Steiger filtering?

Response: Thank you very much for your suggestion. We have incorporated the MR-Steiger filtering analysis into the sensitivity analysis¹². We repeated the analyses after MR-Steiger filtering which removes SNPs suggestive of a reversed causal direction (i.e., those explaining more variance in the outcome than in exposure). Using MR-Steiger filtering method, the exposure of self-reported sleep duration, self-reported short sleep and daytime napping removed SNPs suggestive of reverse causation. Subsequent MR analyses excluding these SNPs showed similar results to the primary analysis (**Table R4** and *Supplementary Table S4*). We have added the description of method and result of the MR-Steiger filtering in the manuscript:

1. lines 278-280, page 13: “Additionally, MR-Steiger filtering was performed to remove reverse causal genetic IVs, which are SNPs that explain more variance in the outcome than in the exposure¹². After removing reverse causal SNPs, we repeated the MR analyses.”
2. lines 339-342, page 16: “Using MR-Steiger filtering method, the exposure of self-reported sleep duration, self-reported short sleep and daytime napping removed SNPs suggestive of reverse causation. Subsequent MR analyses excluding these SNPs showed similar results to the primary analysis (*Supplementary Table S4*).”

Table R4. Mendelian randomization results after Steiger filtering.

Exposure	Methods	Beta (95% CI)	P-value
Self-reported sleep duration	IVW	0.005 (-0.049, 0.060)	0.846
	MR-Egger	-0.036 (-0.243, 0.171)	0.734
	Weighted median	-0.019 (-0.079, 0.041)	0.539
	MR-PRESSO	0.005 (-0.051, 0.039)	0.847
Self-reported short sleep	IVW	-0.024 (-0.363, -0.085)	0.002
	MR-Egger	0.355 (-0.639, 1.350)	0.492

	Weighted median	-0.040 (-0.243, 0.164)	0.702
	MR-PRESSO	-0.224 (-0.441, -0.007)	0.039
Daytime napping			
	IVW	-0.069 (-0.136, 0.003)	0.079
	MR-Egger	-0.146 (-0.378, 0.086)	0.219
	Weighted median	-0.095 (-0.172, -0.019)	0.034
	MR-PRESSO	-0.070 (-0.141, 0.003)	0.062

- Figure 3: not sure what is presented here. Why not presenting leave-one out plots?

Response: Thank you for your comment. Leave-one-out analysis involves iteratively removing one SNP at a time and conducting IVW using the remaining SNPs. **Figure 3** consists of 11 boxplots (**Figure R4** in the letter). Boxplots are graphical representations that display the distribution of a dataset. They show the median, quartiles, and potential outliers of the data. Each boxplot represents the centralized tendency of effect sizes (β coefficients) of sleep-related traits on LTL based on the results of leave-one-out analysis. These boxplots aim to identify outlying SNPs which could potentially bias our effect estimates. For example, if the result was driven by one or a few SNPs with larger effects, then we would expect a drastic change in the estimate when that particular outlying SNP was removed. The points outside the whiskers are potential outliers. We have provided the leave-one-out plot in **Figure R5** in the letter. Compared with leave-one-out plot (**Figure R5**), **Figure R4** could view the potential outliers more visually and aesthetically (especially for y-axis). In order to better understand **Figure R4**, we have revised the footnote of this figure, as follow (lines 774-776, page 39):

“Each boxplot represents the centralized tendency of effect sizes (β coefficients) of A) overall sleep duration, B) short sleep duration, C) long sleep duration, D) insomnia, E) chronotype, F) L5 time, G) sleep episodes, H) sleep efficiency, I) accelerometer-based sleep duration, J) daytime sleepiness, K) daytime napping on LTL based on the results of leave-one-out analysis where we excluded one SNP at a time and performed IVW using the remaining SNPs. The points outside the whiskers are potential outlying SNPs.”

We have also included leave-one-out plot in the Supplementary materials (**Supplementary Figure S2**) and cited the figure in the main text, as follow (lines 336-339, page 16):

“In the leave-one-out analysis, no apparent outlying SNPs were observed, and the results were not influenced by any outlier and clustered closely around the expected value of estimation (Figure 3 and Supplementary Figure S2).”

Figure R4. Sensitivity analysis leaving one SNP out at a time for the association between sleep-related traits and LTL.

Each boxplot represents the centralized tendency of effect sizes (β coefficients) of A) overall sleep duration, B) short sleep duration, C) long sleep duration, D) insomnia, E) chronotype, F) L5 time, G) sleep episodes, H) sleep efficiency, I) accelerometer-based sleep duration, J) daytime sleepiness, K) daytime napping on LTL based on the results of leave-one-out analysis where we excluded one SNP at a time and performed IVW using the remaining SNPs. The points outside the whiskers are potential outlying SNPs.

Figure R5. Plots of leave-one-out analyses for MR analyses of A) self-reported sleep duration, B) self-reported short sleep, C) self-reported long sleep, D) insomnia, E) morning chronotype, F) L5 timing, G) No. of sleep episodes, H) sleep efficiency, I) accelerometer-based sleep duration, J) daytime sleepiness, and K) daytime napping on LTL.

Abbreviations: L5 timing, least active 5 hours timing; LTL, leukocyte telomere length.

The red lines represent the relationships of A) self-reported sleep duration, B) self-reported short sleep, C) self-reported long sleep, D) insomnia, E) morning chronotype, F) L5 timing, G) No. of sleep episodes, H) sleep efficiency, I) accelerometer-based sleep duration, J) daytime sleepiness, and K) daytime napping with LTL using inverse-variance weighted (IVW) method.

Reviewer #3

COMMENTS TO AUTHOR(S)

The study related eleven sleep traits to leukocyte telomere length (LTL) using univariable and multivariable MR. It found a strong causal relationship between short sleep and reduced LTL, while a suggestive causal effect between morning chronotype and longer LTL.

Here, I have some comments.

Response: Thank you very much for your encouraging comments and constructive suggestions. We have addressed all the comments point by point and made changes and improvements in our revised manuscript accordingly. Specifically, we have i) estimate type 1 error rate and bias and performed MRlap analysis to consider the impact of bias introduced by sample overlap on our findings; ii) simplified the description of the summary statistics in the section of “Method”; and iii) provided conditional F -statistics for exposures in MVMR analysis. All changes in the manuscript are tracked in the marked-up version and the line numbers in the following responses refer to the “marked-up” version of the manuscript. We hope that these revisions demonstrate our commitment to addressing your feedback and contribute to making the study more fitting for publication.

1. ‘In this study, we aimed to comprehensively investigate the causal associations between sleep traits and LTL by conducting two-sample MR analyses on a set of 7 self-reported and 4 accelerometer-based sleep traits.’ Self-report bias might exist for some variables. The authors should consider it.

Response: Thank you for sharing your insight. In this study, self-reported sleep traits encompass overall sleep duration, insomnia, Chronotype, daytime sleepiness and napping. Of these, it is prudent to acknowledge that self-report bias could primarily impact the precision of reported sleep duration. This potential bias arises from participants being asked to self-report the number of hours they typically sleep within a 24-hour period, a measurement prone to subjective perceptions and reporting tendencies. However, our analysis incorporated four sleep traits measured through accelerometers: sleep duration, L5 timing, number of sleep episodes, and sleep efficiency. Derived from accelerometers, these objective measurements offer a potentially more dependable reflection of sleep patterns. Additionally, other self-reported sleep traits are integral to daily life, so they might exhibit less susceptibility to significant biases. While we are attentive to the potential impact of self-report bias, it does not materially affect the results according to previous studies^{8,13,14}. For example, the reference article also mentioned that: “*Because participants would not know their value for the genetic IV, any such error would be expected to be random in relation to our outcomes, which would be expected to bias toward the null.*”¹³. Hence, such subtle bias appears to be acceptable.

2. As to the data, all exposure and outcome are from summary statistics online. The study has a detailed description of the definition and processing of variables. However, this could be placed in the supplementary file and replaced with a table.

Response: Thank you very much for your suggestion. We agree with your suggestion. We have provided **Table 1** to summarize the information of the used GWASs and simplified the description of the summary statistics in the section of “Method”. We have

excluded the sentences detailing the process of conducting GWAS for exposures and outcomes, while maintaining the definitions of sleep-related traits and LTL ensure the readability of the manuscript, as follow (lines 149-227, pages 8-11):

“Detailed information of these GWASs was presented in Table 1.

Overall sleep duration was evaluated through self-reporting by asking participants how many hours of sleep they typically get in a 24-hour period (including naps), with only integer values accepted as an answer (N=446,118) ⁴. Additionally, binary variables were created to distinguish short sleep duration (≤ 6 hours vs. 7-8 hours) and long sleep duration (≥ 9 hours vs. 7-8 hours). Among the individuals included in the study, 106,192 reported short sleep duration (≤ 6 hours), 34,184 reported long sleep duration (≥ 9 hours), and 305,742 reported a sleep duration of 7-8 hours. Participants who reported using any sleep medication and those with extreme self-reported sleep durations (< 3 hours or > 18 hours) were excluded from the study.

Insomnia was assessed by self-reported responses to a question that inquired about difficulties falling asleep at night or waking up in the middle of the night. Participants who responded “usually” were classified as having frequent insomnia (N=129,270), whereas those who responded “never/rarely” were considered as controls (N=237,627).

Chronotype refers to an individual’s natural inclination toward being a morning person, an evening person, or somewhere in between ¹⁵. Morning chronotype was assessed in the UK Biobank using self-reported measurement by answering the question of whether they were morning or evening people. Participants who reported “Definitely an ‘evening’ person” and “More an ‘evening’ than ‘morning’ person” were classified as controls (N=150,908) and those who reported “Definitely a ‘morning’ person” and “More a ‘morning’ than ‘evening’ person” were classified as cases (N=252,287). BOLT-LMM with adjustment for age, sex, study center, and genotyping array was performed to estimate the genetic associations between SNPs and morning chronotype ¹⁵. To transform the linear scale of BOLT-LMM association statistics into log odds ratio for the genetic associations with morning chronotype, we utilized the approximation $\log OR \approx \beta / (\mu(1-\mu))$, where β represents the reported effect size from the BOLT-LMM and μ is the case fraction of the binary trait ($\mu=62.6\%$) ¹⁶.

Summary statistics for daytime sleepiness (N=452,071) and napping (N=452,633) were both obtained from the UK Biobank ^{17,18}. Daytime sleepiness and napping were ascertained by asking the questions “How likely are you to dose off or fall asleep during the daytime when you don’t mean to? (e.g. when working, reading or driving)” and “Do you have a nap during the day?”, respectively.

Summary-level data for accelerometer-based sleep traits were obtained from ~85,000 individuals of European ancestry from the UK Biobank ¹⁹. Our analysis was centered on 4 sleep traits measured via accelerometers, referring to L5 timing, number of sleep episodes, sleep duration, and sleep efficiency. One Sleep episode within the sleep period time window (SPT-window) was defined as periods of at least 5 min with a change $< 5^\circ$ associated with the z-axis. The summed duration of all sleep episodes was calculated to define accelerometer-base sleep duration (N=84,810). L5 timing (N=85,205) was defined as the midpoint of the 5h period with the minimum average acceleration of each day. The number of sleep episodes (N=84,441) was defined as the number of sleep episodes within the SPT-window. Sleep efficiency (N=84,810) was calculated as sleep

duration divided by the duration of the SPT-window.

Outcomes The present study utilized the summary statistics for LTL, which were derived from a GWAS meta-analysis conducted on a large sample of 472,174 individuals of European ancestry in the UK Biobank ²⁰. LTL was quantified using the multiplex quantitative polymerase chain reaction (qPCR) methodology, which measures the ratio of telomere repeat copy number (T) relative to a single copy gene, providing a reliable and precise assessment of LTL ²¹. In this process, LTL measurements were log-transformed and Z-standardized to minimize the impact of measurement variability on the results.”

3. The problem of most concern is the overlap of the sample. Even though the instruments are strong, bias relies on the covariance of associations between the variant–exposure and variant-outcome in summarized data. The methods to assess bias from sample overlap is still not convincing, more ways of assessment can be considered (e.g., <https://sb452.shinyapps.io/overlap/>). However, why did the authors use the definitely UKB samples as an outcome, rather than another, e.g., FinnGen?

Response: Thank you very much for your comment. We understand the importance of utilizing such non-overlapped exposure and outcome GWAS datasets to avoid potential bias. However, after a comprehensive search in PubMed, web of science, and GWAS catalogue, we have encountered challenges in finding suitable non-overlapping GWAS datasets that align with our research focus. For sleep-related traits, although we need the genome-wide significant SNPs, genome-wide significant SNPs were identified using UK Biobank cohort. For LTL, we need the whole summary-level genetic data of LTL, the currently publicly available dataset was analysed using UK Biobank cohort. Given the current limitations in finding non-overlapping datasets, we conducted a series of sensitivity analyses to investigate whether the bias introduced by sample overlap impacted our findings.

As you suggested, we have further assessed the bias and type 1 error rate for sample overlap using the online calculator (<https://sb452.shinyapps.io/overlap/>). As shown in **Table R5**, the type 1 error rate due to sample overlap between sleep-related traits and LTL was controlled under 0.05 and the biases were minimal (<2.5%). Furthermore, we employed a recently developed MR technique known as MRlap to explore the associations between sleep-related traits and LTL ¹. MRlap could account and correct for bias arising from sample overlap, weak instruments and winner’s curse. MRlap using cross-trait LDSC to approximate the sample overlap and combats sample overlap, for example, by reducing an observed 15% overestimation for fully overlapping samples to a 5% overestimation ¹. MRlap estimates were largely similar to the primary analysis (*Supplementary Table S12*). Specifically, genetically predicted self-reported short sleep was associated with decreased LTL (Corrected β [SE]: -0.123 [0.062]; $P=0.047$), while morning chronotype was associated with increased LTL (Corrected β [SE]: 0.024 [0.012]; $P=0.036$; *Supplementary Table S12*). We have revised the description of the methods and results of these sensitivity analyses in the manuscript, as follow:

1. lines 292-297, page 14: “Finally, we employed multiple methods to assess the potential impact of bias introduced by partial sample overlap between sleep traits and LTL on our findings. Specifically, i) we evaluated the bias and type 1 error rate for sample overlap using an online calculator (<https://sb452.shinyapps.io/overlap/>) ², and

ii) we utilized a robust MR method, MRlap, designed to mitigate bias introduced by sample overlap, winner's curse and weak instruments ¹."

2. lines 397-400, page 18: "The type 1 error rate due to the sample overlap between sleep-related traits and LTL was controlled under 0.05 and the biases were minimal (Supplementary Table S11). Additionally, the MRlap analyses showed concordant results with primary results (Supplementary Table S12)."

We have discussed the bias in the limitation (lines 521-530, page 23).

"First, sample overlap of the current two-sample MR study due to both samples of sleep-related traits and LTL being from UK Biobank may lead to bias and type 1 error (false positive) inflation. For this, the type 1 error rate due to sample overlap between sleep-related traits and LTL was controlled under 0.05, and the MR estimates were well validated by using the MRlap method. Furthermore, a recent simulation study provided support for the rationality and validity of two-sample MR using overlapping samples in large cohorts ³."

Table R5. The degree of overlapping between samples of sleep-related traits and samples of leukocyte telomere length.

Exposures	Sample size of exposure GWAS	Outcomes	Sample size of outcome GWAS	Overlap degree	Number of instruments	R^2	Observational estimate	Bias	Type 1 error rate
Chronotype	403,195	LTL	472,174	85.39%	117	1.296%	0.016 ^a	0.0002	0.05
Insomnia	237,627	LTL	472,174	50.33%	39	0.698%	-0.034 ^a	0.0004	0.05
Self-reported sleep duration	446,118	LTL	472,174	94.48%	66	0.604%	0.055 ^a	0.0013	0.05
Self-reported short sleep	411,934	LTL	472,174	87.24%	24	0.219%	-0.315 ^a	-0.0064	0.05
Self-reported long sleep	339,926	LTL	472,174	71.99%	10	0.117%	0.252 ^a	0.0063	0.05
Daytime napping	452,633	LTL	472,174	95.86%	105	1.061%	-0.034 ^a	-0.0012	0.05
Daytime sleepiness	452,071	LTL	472,174	95.74%	37	0.343%	-0.005 ^a	<0.0001	0.05
Accelerometer-based sleep duration	84,810	LTL	472,174	17.96%	9	0.516%	-0.004 ^a	<0.0001	0.05
L5 timing	85,205	LTL	472,174	18.05%	5	0.347%	-0.013 ^a	<0.0001	0.05
Sleep efficiency	84,810	LTL	472,174	17.96%	5	0.267%	-0.02 ^a	-0.0006	0.05
No. of sleep episodes	84,441	LTL	472,174	17.88%	22	0.982%	-0.002 ^a	<0.0001	0.05

Abbreviations: Genome-wide association study (GWAS); leukocyte telomere length (LTL).

^a Since the observational estimates were not available from previous studies, we applied the estimates in our study as the substitute parameter.

4. In using MVMR, did the authors consider using some strategies like in UVMR (Cochran Q test and I2 index) to exclude outliers and check heterogeneity? For F statistics, conditional F-statistic can be presented for MVMR.

Response: We appreciate a lot for your advice. We have calculated the conditional F-statistics for exposures in MVMR analysis and added into the *Supplementary Table S8* and *Supplementary Table S9*. Here, we presented the results of conditional F-statistics in **Table R6**. We have provided the results of Cochran Q test in the MVMR analysis (**Table R6** and *Supplementary Table S8 and S9*). In our study, there is evidence of heterogeneity for both MVMR models (P for Cochran Q test <0.001). To alleviate the impact of heterogeneity, we used MVMR-IVW with random effects model to estimate the effect of the exposures on LTL. Through carefully retrieve the PubMed, Web of Science, and Google scholar, there is no method to calculate I^2 in MVMR method. We have added the sentence in the manuscript (lines 261-263, page 12):

“Heterogeneity in the MVMR analysis was examined by Cochran Q test. For MVMR, conditional F-statistics were calculated for the exposures and larger than 10 indicates a lower risk of weak instrument bias ²².”

By the way, we have updated the summary statistics of smoking and drinking using the Phase 2 of GSCAN dataset (previously used Phase 1 of GSCAN dataset) ⁹. Phase 2 of GSCAN dataset have more samples than phase 1 of GSCAN dataset and make the genetic estimates more accurate. The updated results were provided in **Table R6**. MVMR-IVW method showed the inverse direct effect of self-reported short sleep duration on LTL (-0.159 [-0.310, -0.009]; $P=0.038$) after adjusting for the effect of BMI, smoking, and alcohol consumption (**Table R6**). After adjusting for the effect of self-reported sleep duration, insomnia, BMI, smoking, and alcohol consumption, we observed that MVMR results supported the findings of a positive effect of morning chronotype on LTL (0.017 [0.001, 0.033]; $P=0.048$; **Table R6**).

Table R6. MVMR results.

Exposures	Conditional F-statistics	MVMR-IVW			
		Beta (95% CI)	P-value	Cochran's Q test	P-value for Q test
Model 1				1091.8	<0.001
Self-reported short sleep	13.721	-0.159 (-0.310, -0.009)	0.038		
smoke	13.463	-0.002 (-0.060, 0.057)	0.954		
drink	15.149	-0.048 (-0.114, 0.019)	0.163		
BMI	21.72	-0.025 (-0.049, -0.002)	0.035		
Model 2				673.51	<0.001
Self-reported sleep duration	5.448	-0.036 (-0.087, 0.014)	0.157		
insomnia	10.628	-0.070 (-0.189, 0.049)	0.25		

chronotype	15.526	0.017 (0.001, 0.033)	0.048
BMI	18.956	-0.018 (-0.005, -0.031)	0.006
smoke	14.955	-0.043 (-0.079, -0.008)	0.017
drink	12.255	-0.037 (-0.082, 0.007)	0.102

Reference

- 1 Mounier, N. & Kutalik, Z. Bias correction for inverse variance weighting Mendelian randomization. *Genet Epidemiol* **47**, 314-331, doi:10.1002/gepi.22522 (2023).
- 2 Burgess, S., Davies, N. M. & Thompson, S. G. Bias due to participant overlap in two-sample Mendelian randomization. *Genet Epidemiol* **40**, 597-608, doi:10.1002/gepi.21998 (2016).
- 3 Minelli, C. *et al.* The use of two-sample methods for Mendelian randomization analyses on single large datasets. *Int J Epidemiol* **50**, 1651-1659, doi:10.1093/ije/dyab084 (2021).
- 4 Dashti, H. S. *et al.* Genome-wide association study identifies genetic loci for self-reported habitual sleep duration supported by accelerometer-derived estimates. *Nat Commun* **10**, 1100, doi:10.1038/s41467-019-08917-4 (2019).
- 5 Ayas, N. T. *et al.* A prospective study of self-reported sleep duration and incident diabetes in women. *Diabetes Care* **26**, 380-384 (2003).
- 6 Qureshi, A. I., Giles, W. H., Croft, J. B. & Bliwise, D. L. Habitual sleep patterns and risk for stroke and coronary heart disease: a 10-year follow-up from NHANES I. *Neurology* **48**, 904-911 (1997).
- 7 Cappuccio, F. P., D'Elia, L., Strazzullo, P. & Miller, M. A. Sleep duration and all-cause mortality: a systematic review and meta-analysis of prospective studies. *Sleep* **33**, 585-592 (2010).
- 8 Richmond, R. C. *et al.* Investigating causal relations between sleep traits and risk of breast cancer in women: mendelian randomisation study. *BMJ* **365**, l2327, doi:10.1136/bmj.l2327 (2019).
- 9 Saunders, G. R. B. *et al.* Genetic diversity fuels gene discovery for tobacco and alcohol use. *Nature* **612**, 720-724, doi:10.1038/s41586-022-05477-4 (2022).
- 10 Lawlor, D. A., Harbord, R. M., Sterne, J. A. C., Timpson, N. & Davey Smith, G. Mendelian randomization: using genes as instruments for making causal inferences in epidemiology. *Stat Med* **27**, 1133-1163 (2008).
- 11 Liang, C. *et al.* Infertility, recurrent pregnancy loss, and risk of stroke: pooled analysis of individual patient data of 618 851 women. *BMJ* **377**, e070603, doi:10.1136/bmj-2022-070603 (2022).
- 12 Hemani, G., Tilling, K. & Davey Smith, G. Orienting the causal relationship between imprecisely measured traits using GWAS summary data. *PLoS Genet* **13**, e1007081, doi:10.1371/journal.pgen.1007081 (2017).
- 13 Liu, J. *et al.* Assessing the Causal Role of Sleep Traits on Glycated Hemoglobin: A Mendelian Randomization Study. *Diabetes Care* **45**, 772-781, doi:10.2337/dc21-0089 (2022).
- 14 Ai, S. *et al.* Causal associations of short and long sleep durations with 12 cardiovascular diseases: linear and nonlinear Mendelian randomization

- analyses in UK Biobank. *Eur Heart J* **42**, 3349-3357, doi:10.1093/eurheartj/ehab170 (2021).
- 15 Jones, S. E. *et al.* Genome-wide association analyses of chronotype in 697,828 individuals provides insights into circadian rhythms. *Nat Commun* **10**, 343, doi:10.1038/s41467-018-08259-7 (2019).
- 16 Lloyd-Jones, L. R., Robinson, M. R., Yang, J. & Visscher, P. M. Transformation of Summary Statistics from Linear Mixed Model Association on All-or-None Traits to Odds Ratio. *Genetics* **208**, 1397-1408, doi:10.1534/genetics.117.300360 (2018).
- 17 Wang, H. *et al.* Genome-wide association analysis of self-reported daytime sleepiness identifies 42 loci that suggest biological subtypes. *Nat Commun* **10**, 3503, doi:10.1038/s41467-019-11456-7 (2019).
- 18 Dashti, H. S. *et al.* Genetic determinants of daytime napping and effects on cardiometabolic health. *Nature Communications* **12**, 900, doi:10.1038/s41467-020-20585-3 (2021).
- 19 Jones, S. E. *et al.* Genetic studies of accelerometer-based sleep measures yield new insights into human sleep behaviour. *Nat Commun* **10**, 1585, doi:10.1038/s41467-019-09576-1 (2019).
- 20 Codd, V. *et al.* Polygenic basis and biomedical consequences of telomere length variation. *Nat Genet* **53**, 1425-1433, doi:10.1038/s41588-021-00944-6 (2021).
- 21 Cawthon, R. M. Telomere length measurement by a novel monochrome multiplex quantitative PCR method. *Nucleic Acids Res* **37**, e21, doi:10.1093/nar/gkn1027 (2009).
- 22 Sanderson, E., Spiller, W. & Bowden, J. Testing and correcting for weak and pleiotropic instruments in two-sample multivariable Mendelian randomization. *Stat Med* **40**, 5434-5452, doi:10.1002/sim.9133 (2021).

Reviewers' comments:

Reviewer #1 (Remarks to the Author):

Thank you for taking the time to respond and address my comments so carefully. I am satisfied with the amendments and have no further points to raise.

Reviewer #2 (Remarks to the Author):

The authors addressed my concerns. Interesting paper!

Reviewer #3 (Remarks to the Author):

The authors almost addressed my questions with detailed description and tables. However, it did not seem the authors had heterogeneity and weak in MVMR. Actually, there are published workflow to address the pleiotropic SNPs. For example, in UMR, besides MRPRESSO, Cochran's Q and Rucker's Q can help exclude outliers. In MVMR, MRPRESSO and MRLASSO, etc, are also usable. Authors can learn from e.g., RadialMR and Q-minimisation in weak instrument. Results of MVMR can be more robust and convincing using these methods.

Reviewers' Comments to Author:

Reviewer #1

COMMENTS TO AUTHOR(S)

Thank you for taking the time to respond and address my comments so carefully. I am satisfied with the amendments and have no further points to raise.

Response: Thank you for your kind words and for your thorough review of our manuscript. We genuinely appreciate your time and effort in providing valuable feedback. On behalf my co-authors, we would like to express our great appreciation to your recognition.

Reviewer #2

COMMENTS TO AUTHOR(S)

The authors addressed my concerns. Interesting paper!

Response: We sincerely appreciate the time and effort that you dedicated to providing positive feedback and insightful comments on our manuscript. We truly appreciate your valuable feedback, which has been instrumental in improving the quality of our work. We are delighted to hear that you are satisfied with the amendments we made based on your suggestions.

Reviewer #3

COMMENTS TO AUTHOR(S)

The authors almost addressed my questions with detailed description and tables. However, it did not seem the authors had heterogeneity and weak in MVMR. Actually, there are published workflow to address the pleiotropic SNPs. For example, in UMR, besides MRPRESSO, Cochran's Q and Rucker's Q can help exclude outliers. In MVMR, MRPRESSO and MRLASSO, etc, are also usable. Authors can learn from e.g., RadialMR and Q -minimisation in weak instrument. Results of MVMR can be more robust and convincing using these methods.

Response: We appreciate you a lot for your encouraging comments and insightful suggestions. All changes in the manuscript are tracked in the marked-up version and the line numbers in the following responses refer to the "marked-up" version of the manuscript. Following your suggestions, we have carefully read the paper about radial MR¹ and Q -minimisation in weak instrument². Subsequently, we have revised the manuscript thoroughly and conducted additional analyses. Briefly, we complemented radial MR analysis to identify outliers and repeated the univariable MR (UVMR) analysis without these SNPs as sensitivity analysis. Furthermore, we complemented MVMR-LASSO, MVMR-PRESSO, and MVMR- Q (het) methods as sensitivity MVMR methods. Detailed descriptions and results were described below.

First, we performed radial MR-IVW and radial MR-Egger analyses in the UVMR analyses to identify outliers and repeated the UVMR analysis without these SNPs as sensitivity analysis¹. Briefly, Radial MR identified outliers with the more weight in the MR analysis and the large contribution to Cochran's Q statistic (for radial IVW) or Rucker's Q statistic (for radial MR-Egger) for heterogeneity. Radial MR analysis was conducted using modified second order weights and a P -value threshold (0.05 divided by the number of SNPs) for identifying outliers. We performed the radial MR analysis using "RadialMR" package (version 1.1) in R software. For the UVMR analyses of 11 sleep-related traits on LTL, radial MR identified one to twenty outliers (**Figure R1**).

Also, we have incorporated the radial plots of 11 sleep traits on LTL into the Supplementary materials as *Supplementary Figure S3*. Then, after excluding the outliers, Cochran's Q test and Rucker's Q test showed no evidence of heterogeneity of these analyses (all P -values >0.05 ; **Table R1**). The primary UVMR analysis (i.e., IVW method) yielded similar results, indicating genetically predicted self-reported short sleep was associated with decreased LTL (β [95% CI]: -0.155 [-0.302, -0.007]; $P=0.040$) and genetically predicted morning chronotype was associated with increased LTL (0.012 [0.001, 0.024]; $P=0.041$; **Table R1**). Other sensitivity MR methods did not alter the results markedly (**Table R1**). We have included the results of re-run UVMR analyses without these outliers as *Supplementary Table S4* in the supplementary material. We also have added the complementary analysis into the manuscript:

1. Lines 237-241, page 11: "*Additionally, we performed radial MR-IVW and radial MR-Egger analyses using modified second-order weights to identify outliers and repeated the UVMR analyses without these outliers as sensitivity analyses¹. Briefly, radial MR-IVW and radial MR-Egger analyses identify outliers that make large contributions to Cochran's Q statistic or Rucker's Q statistic for heterogeneity, respectively.*"
2. Lines 273-274, page 12: "*All analyses were conducted using various R software (version 4.2.2) packages including TwoSampleMR (version 0.5.6), MVMR (version 0.3), coloc (version 5.1.0.1), locuscomparer (version 1.0.0), MR-PRESSO (version 1.0), MRlap (version 0.0.3.0), MendelianRandomization (version 0.7.0), and RadialMR (version 1.1), with a two-sided approach.*"
3. Lines 301-304, page 13: "*Radial MR analyses identified one to twenty outliers for these UVMR analyses of 11 sleep-related traits on LTL (Supplementary Figure S3). After excluding the outliers detected by radial MR analyses, the results did not change markedly (Supplementary Table S4).*"

Second, MVMR-LASSO, MVMR-PRESSO, and MVMR- Q (het) methods were additionally used as sensitivity MVMR methods to enhance the robustness and convincement of MVMR results^{2,3}. MVMR-LASSO provides the best estimation in terms of mean squared error for moderate to high levels of pleiotropy³. MVMR-PRESSO performs a test based on a heterogeneity measure to identify outliers, which are then removed from the analysis. After removing the outliers, we re-estimated the effects. MVMR- Q (het) offers estimations that are robust to the impact of weak instruments by incorporating the covariance matrix of the phenotypic associations to adjust the weights. MVMR- Q (het) method minimizes the Q -statistic and allows for heterogeneity². Because our study is two-sample MR design, the covariance matrix of the phenotypic associations is not available. We used genetic correlation matrix of the phenotypes as the substitution of the phenotypic correlation matrix. And we specified 100 bootstrap iterations to calculate the 95% CI of the effects. Above MVMR analyses were implemented using the following R packages: MendelianRandomization ('mr_mvlasso' function for MVMR-LASSO), MVMR ('qhet_mvmr' function for MVMR- Q (het)), and MRPRESSO ('mr_presso' function for MVMR-PRESSO). These MVMR methods showed similar results with the results of previous MVMR-IVW and MVMR-Egger methods in the manuscript. After adjusting for the effect of self-reported sleep duration, insomnia, BMI, smoking, and alcohol consumption, MVMR-LASSO, MVMR-PRESSO, and MVMR- Q (het) methods provided the evidence of positive effect of morning chronotype on LTL (**Table R2**). MVMR-LASSO, MVMR-PRESSO, and MVMR- Q (het) supported the evidence of inverse direct effect of self-reported short sleep duration on LTL after adjusting for the effect of BMI, smoking, and alcohol

consumption (**Table R2**). The results of these MVMR methods were also included in the *Supplementary Table S9* and *S10*. Also, we have incorporated the complementary MVMR analyses into the manuscript:

1. Lines 253-257, page 11: “*Fifth, we conducted sensitivity analyses of MVMR that are robust to pleiotropy, including MVMR-Egger, MVMR-PRESSO, and MVMR-least absolute shrinkage and selection operator (MVMR-LASSO)³. Furthermore, we conducted MVMR-Q(het) method, which aimed to minimize the Q-statistics and allow for heterogeneity².*”
2. Lines 345-347, page 15: “*These findings remained consistent in the MVMR-LASSO, MVMR-PRESSO, and MVMR-Q(het) methods (Supplementary Table S9-S10).*”

After complementing these analyses, the findings of the manuscript did not alter. Consequently, we believe our conclusion is reliable.

- IW and MR-Egger Outlier
- MR-Egger Outlier
- IW Outlier
- IW
- MR-Egger

Figure R1. Radial Mendelian randomization plots of A) self-reported sleep duration, B) self-reported short sleep, C) self-reported long sleep, D) insomnia, E) morning chronotype, F) L5 timing, G) No. of sleep episodes, H) sleep efficiency, I) accelerometer-based sleep duration, J) daytime sleepiness, and K) daytime napping on LTL.

Abbreviations: IVW, inverse-variance weighted; L5 timing, least active 5 hours timing; LTL, leukocyte telomere length.

Radial MR method identify outliers with the most weight in the MR analysis and the largest contribution to Cochran's Q statistic (for radial IVW) or Rucker's Q statistic (for radial Egger) for heterogeneity, which may then be removed and the data re-analyzed.

Radial curve displays the ratio estimate for the outliers (identified using radial MR), as well as the radial IVW (in blue) and radial MR-Egger regression (in orange).

Table R1. MR results of 11 sleep-related traits on leukocyte telomere length after removing outliers (identified using Radial MR).

Methods	No. of SNPs	β	95% CI	P
Chronotype				
IVW	102	0.012	0.001 to 0.024	0.041
Cochran's Q test	102	/	/	0.820
MR-Egger	102	0.029	-0.012 to 0.071	0.17
MR-Egger intercept test	102	/	/	0.396
Rucker's Q test	102	/	/	0.815
Weighted median	102	0.02	0.001 to 0.039	0.039
MR-PRESSO	102	0.012	0.001 to 0.024	0.044
MR-PRESSO Global test	102	/	/	0.825
Insomnia				
IVW	28	-0.002	-0.085 to 0.080	0.959
Cochran's Q test	28	/	/	0.277
MR-Egger	28	-0.165	-0.425 to 0.095	0.223
MR-Egger intercept test	28	/	/	0.203
Rucker's Q test	28	/	/	0.313
Weighted median	28	-0.011	-0.134 to 0.112	0.862
MR-PRESSO	28	-0.002	-0.090 to 0.086	0.962
MR-PRESSO Global test	28	/	/	0.272
Self-reported sleep duration				
IVW	54	0.014	-0.027 to 0.055	0.507
Cochran's Q test	54	/	/	0.094
MR-Egger	54	0.035	-0.116 to 0.186	0.654
MR-Egger intercept test	54	/	/	0.781
Rucker's Q test	54	/	/	0.081
Weighted median	54	-0.018	-0.076 to 0.039	0.532
MR-PRESSO	54	0.014	-0.027 to 0.055	0.51
MR-PRESSO Global test	54			0.096
Self-reported short sleep				
IVW	20	-0.155	-0.302 to -0.007	0.040
Cochran's Q test	20	/	/	0.161
MR-Egger	20	-0.035	-0.834 to 0.764	0.458
MR-Egger intercept test	20	/	/	0.932
Rucker's Q test	20	/	/	0.134
Weighted median	20	-0.036	-0.250 to 0.177	0.739
MR-PRESSO	20	-0.155	-0.309 to -0.001	0.049
MR-PRESSO Global test	20			0.162
Self-reported long sleep				
IVW	8	0.34	-0.072 to 0.752	0.106
Cochran's Q test	8	/	/	0.423
MR-Egger	8	-1.746	-4.332 to 0.840	0.234
MR-Egger intercept test	8	/	/	0.160
Rucker's Q test	8	/	/	0.610
Weighted median	8	0.09	-0.438 to 0.618	0.738
MR-PRESSO	8	0.34	-0.074 to 0.754	0.151
MR-PRESSO Global test	8	/	/	0.413

Daytime napping				
IVW	85	-0.066	-0.137 to 0.005	0.068
Cochran's Q test	85	/	/	0.926
MR-Egger	85	-0.068	-0.239 to 0.103	0.438
MR-Egger intercept test	85	/	/	0.980
Rucker's Q test	85	/	/	0.914
Weighted median	85	-0.094	-0.171 to -0.017	0.016
MR-PRESSO	85	-0.066	-0.141 to 0.010	0.085
MR-PRESSO Global test	85	/	/	0.932
Daytime sleepiness				
IVW	28	-0.039	-0.149 to 0.071	0.487
Cochran's Q test	28	/	/	0.756
MR-Egger	28	-0.01	-0.471 to 0.450	0.965
MR-Egger intercept test	28	/	/	0.900
Rucker's Q test	28	/	/	0.710
Weighted median	28	-0.007	-0.167 to 0.153	0.933
MR-PRESSO	28	-0.039	-0.138 to 0.060	0.444
MR-PRESSO Global test	28	/	/	0.766
Accelerometer-based sleep duration				
IVW	8	0.01	-0.030 to 0.050	0.613
Cochran's Q test	8	/	/	0.352
MR-Egger	8	0.074	-0.027 to 0.175	0.201
MR-Egger intercept test	8	/	/	0.227
Rucker's Q test	8	/	/	0.426
Weighted median	8	0.002	-0.053 to 0.056	0.201
MR-PRESSO	8	0.01	-0.032 to 0.053	0.646
MR-PRESSO Global test	8	/	/	0.361
L5 timing				
IVW	4	-0.028	-0.077 to 0.020	0.254
Cochran's Q test	4	/	/	0.669
MR-Egger	4	-0.069	-0.153 to 0.016	0.251
MR-Egger intercept test	4	/	/	0.369
Rucker's Q test	4	/	/	0.888
Weighted median	4	-0.027	-0.082 to 0.029	0.343
MR-PRESSO	4	-0.028	-0.063 to 0.007	0.212
MR-PRESSO Global test	4	/	/	0.629
Sleep efficiency*				
IVW	2	0.026	-0.075 to 0.126	0.614
Cochran's Q test	2			0.147
No. of sleep episodes				
IVW	18	-0.015	-0.046 to 0.016	0.347
Cochran's Q test	18	/	/	0.489
MR-Egger	18	0.013	-0.121 to 0.146	0.854
MR-Egger intercept test	18	/	/	0.679
Rucker's Q test	18	/	/	0.432
Weighted median	18	-0.017	-0.061 to 0.026	0.437
MR-PRESSO	18	-0.015	-0.046 to 0.016	0.353
MR-PRESSO Global test	18	/	/	0.481

Abbreviations: inverse variance weighted (IVW); least active 5 hours timing (L5 timing); Mendelian randomisation (MR); MR-Pleiotropy RESidual Sum and Outliers (MR-PRESSO); single nucleotide polymorphism (SNP).

Because the number of SNPs for sleep efficiency is 2, only IVW method was applicable. MR-Egger method, with an intercept term differing from zero representing evidence of horizontal pleiotropy, the intercept with P -value < 0.05 indicates the presence of overall horizontal pleiotropy.

Cochran's Q statistic in IVW methods to assess heterogeneity of the Wald estimates across variants, P -value < 0.05 indicates the presence of heterogeneity.

Rucker's Q is an extended version of Cochran's Q statistic and can be used to assess heterogeneity about the MR-Egger fit, P -value < 0.05 indicates the presence of heterogeneity.

MR-PRESSO Global test detected horizontal pleiotropy. If the P -values of the global test was less than 0.05, which indicates the existence of horizontal pleiotropy.

Table R2. Results of MVMR models.

Exposures	MVMR-IVW			MVMR-Egger			MVMR-LASSO			MVMR-PRESSO			MVMR-Q(het)	
	β	95% CI	P	β	95% CI	P	β	95% CI	P	β	95% CI	P	β	95% CI
MVMR model 1														
Self-reported short sleep	-0.159	-0.310 to -0.009	0.038	-0.216	-0.430 to -0.001	0.048	-0.102	-0.201 to -0.004	0.043	-0.107	-0.207 to -0.007	0.036	-0.139	-0.275 to -0.003
smoke	-0.002	-0.060 to 0.057	0.954	-0.002	-0.060 to 0.056	0.951	0.003	-0.038 to 0.044	0.889	-0.01	-0.059 to 0.039	0.698	0.011	-0.052 to 0.074
drink	-0.048	-0.114 to 0.019	0.163	-0.048	-0.115 to 0.019	0.158	-0.038	-0.089 to 0.013	0.142	-0.026	-0.084 to 0.033	0.392	-0.066	-0.142 to 0.010
BMI	-0.025	-0.049 to -0.002	0.035	-0.027	-0.051 to -0.003	0.028	-0.031	-0.051 to -0.003	2.47E-04	-0.031	-0.051 to -0.011	0.003	-0.04	-0.059 to -0.021
MVMR model 2														
Self-reported sleep duration	-0.036	-0.087 to 0.014	0.157	0.003	-0.076 to 0.083	0.938	-0.049	-0.108 to 0.009	0.097	-0.025	-0.076 to 0.026	0.333	0.018	-0.033 to 0.070
insomnia	-0.07	-0.189 to 0.049	0.25	-0.087	-0.215 to 0.041	0.183	-0.107	-0.238 to 0.024	0.109	-0.047	-0.155 to 0.062	0.398	-0.039	-0.097 to 0.019
chronotype	0.017	0.001 to 0.033	0.048	0.015	0.001 to 0.029	0.049	0.015	0.001 to 0.028	0.032	0.017	0.001 to 0.033	0.049	0.022	0.002 to 0.041
BMI	-0.018	-0.005 to -0.031	0.006	-0.023	-0.006 to -0.040	0.008	-0.046	-0.073 to -0.018	0.001	-0.059	-0.100 to -0.017	0.006	-0.042	-0.079 to -0.005
smoke	-0.043	-0.079 to -0.008	0.017	-0.041	-0.078 to -0.003	0.033	-0.051	-0.088 to -0.014	0.007	-0.042	-0.073 to -0.010	0.009	-0.029	-0.048 to -0.009
drink	-0.037	-0.082 to 0.007	0.102	-0.049	-0.096 to -0.002	0.042	-0.031	-0.082 to 0.021	0.243	-0.036	-0.077 to 0.004	0.082	-0.056	-0.101 to -0.011

Abbreviations: multivariable Mendelian randomization (MVMR); inverse-variance weighted (IVW); LASSO, least absolute shrinkage and selection operator.

MVMR-PRESSO method returned outliers-corrected estimates.

Reference

- 1 Bowden, J. *et al.* Improving the visualization, interpretation and analysis of two-sample summary data Mendelian randomization via the Radial plot and Radial regression. *Int J Epidemiol* **47**, 1264-1278, doi:10.1093/ije/dyy101 (2018).
- 2 Sanderson, E., Spiller, W. & Bowden, J. Testing and correcting for weak and pleiotropic instruments in two-sample multivariable Mendelian randomization. *Stat Med* **40**, 5434-5452, doi:10.1002/sim.9133 (2021).
- 3 Grant, A. J. & Burgess, S. Pleiotropy robust methods for multivariable Mendelian randomization. *Stat Med* **40**, 5813-5830, doi:10.1002/sim.9156 (2021).

REVIEWERS' COMMENTS:

Reviewer #3 (Remarks to the Author):

The authors addressed all of my questions.

Reviewers' Comments to Author:

Reviewer #3

COMMENTS TO AUTHOR(S)

The authors addressed all of my questions.

Response: Thank you for your kind words and for your thorough review of our manuscript. We genuinely appreciate your time and effort in providing valuable feedback. On behalf my co-authors, we would like to express our great appreciation to your recognition.